# Sketch-Augmented Features Improve Learning Long-Range Dependencies in Graph Neural Networks

**Ryien Hosseini**[1], **Filippo Simini**[2], **Venkatram Vishwanath**[2],
**Rebecca Willett**[1,3], **Henry Hoffmann**[1]
[1]University of Chicago   [2]Argonne National Laboratory
[3] NSF-Simons National Institute for Theory and Mathematics in Biology
{ryien,willett,hankhoffmann}@uchicago.edu
{fsimini,venkat}@anl.gov

## Abstract

Graph Neural Networks learn on graph-structured data by iteratively aggregating local neighborhood information. While this local message passing paradigm imparts a powerful inductive bias and exploits graph sparsity, it also yields three key challenges: (i) oversquashing of long-range information, (ii) oversmoothing of node representations, and (iii) limited expressive power. In this work we inject randomized global embeddings of node features, which we term *Sketched Random Features*, into standard GNNs, enabling them to efficiently capture long-range dependencies. The embeddings are unique, distance-sensitive, and topology-agnostic—properties which we analytically and empirically show alleviate the aforementioned limitations when injected into GNNs. Experimental results on real-world graph learning tasks confirm that this strategy consistently improves performance over baseline GNNs, offering both a standalone solution and a complementary enhancement to existing techniques such as graph positional encodings.

## 1   Introduction

Graph Neural Networks (GNNs) are a widely adopted approach for representation learning on graph-structured data. Standard GNN architectures employ a message-passing framework [26], wherein each node's features are iteratively propagated and aggregated along edges. Thus, GNNs leverage a strong topology-induced bias: node features are iteratively refined by their neighbors, enabling local interactions to shape final node or graph embeddings. Moreover, these localized computations naturally exploit graph sparsity, facilitating efficient training. Despite this success, three persistent challenges remain:

- **Oversquashing:** Signals from distant nodes are compressed into a fixed size vector when traversing multiple hops, hindering a GNN's ability to capture long-range interactions [2].
- **Oversmoothing:** As network depth grows, node representations converge exponentially to nearly identical values, eliminating meaningful distinctions [44, 56].
- **Limited Expressive Power:** Standard GNNs have expressive power no greater than the 1-dimensional Weisfeiler–Lehman heuristic [67], leading them to fail at distinguishing many non-isomorphic graphs or structurally distinct subgraphs [68].

To address these limitations, two primary lines of research have emerged. The first adapts transformers [65] for graph-structured data, either by combining message passing with dense attention layers [59, 70, 38, 46] or by replacing message passing altogether [39, 49]. This approach allows the model to attend to arbitrary node pairs regardless of geodesic distance, thus circumventing many limitations of message-passing GNNs. However, these methods typically incur $O(N^2)$ memory and computation cost in the number of nodes $N$ and rely on carefully designed node-level signals (called *positional*

39th Conference on Neural Information Processing Systems (NeurIPS 2025).

or *structural encodings*) to inject graph structure back into the model [39]. These encodings can be challenging to design, and their theoretical properties remain only partially understood [63].

A second line of work augments traditional GNNs with encodings—often inspired by transformers— to improve performance by enhancing expressiveness [53]. These include purely random node features that break symmetry and guarantee universal separation of non-isomorphic graphs [1, 62]. However, such unstructured noise can slow convergence or degrade learning in practice [7]. In contrast, *structural encodings* (e.g., spectral embeddings [23]) are topologically distance-sensitive but rely on deterministic, graph-based representations that are difficult to make both *unique* and *equivariant* under node permutation. This is because, unlike Euclidean spaces, graphs lack a canonical coordinate system that defines cardinal direction, creating ambiguities that the GNN must handle. Notably, both these approaches primarily focus on topology-derived encodings, and do not consider informative node features typically present in real-world graphs.

This observation motivates our approach: *instead of abandoning message passing or augmenting it solely with structural information, we propose leveraging node features themselves to address GNN limitations.* To this end, we introduce **Sketched Random Features (SRF)**, presented in Algorithm 1, a simple yet powerful method for injecting global, feature-distance-sensitive signals into GNNs. SRF augments each node's features with *sketches*, or randomized low-dimensional projections, of all node features in a kernel space, preserving feature similarity in expectation and breaking symmetry through randomization. This paper describes how sketched random features can augment feature representations in every layer of a message passing GNN to yield accurate predictions in a variety of settings. By the Johnson–Lindenstrauss lemma [35, 8], such projections preserve important distance relationships in feature space even in significantly reduced dimensions. We leverage these properties to construct node-feature representations that we show overcome the challenges above.

Consider a graph's node feature matrix $X \in \mathbb{R}^{N \times F}$, where the $i$-th column of $X^T$, $\mathbf{x}_i \in \mathbb{R}^F$ denotes the features of node $i$, and a positive-definite similarity function (kernel) $\kappa : \mathbb{R}^F \times \mathbb{R}^F \to \mathbb{R}$ defined between these features. Let $\mathcal{E} : \mathbb{R}^{N \times F} \to \mathbb{R}^{N \times D}$ denote a mapping that embeds $X$ into a random feature space via feature map $\varphi(\cdot)$.

While *any* random kernel feature $\varphi(\mathbf{x}_i) \in \mathbb{R}^D$ (e.g., via random Fourier features [57]) already provides an unbiased approximation of similarity between node features, i.e., $\mathbb{E}\big[\varphi(\mathbf{x}_i)^\top \varphi(\mathbf{x}_j)\big] = \kappa(\mathbf{x}_i, \mathbf{x}_j)$, we go further by applying a *cross-node random projection* $\mathcal{S}^{(k)} \in \mathbb{R}^{N \times N}$. Let $\Phi \in \mathbb{R}^{N \times D}$ (with $(D \ll N)$ be the matrix of random kernel features, where the $i$-th row of $\Phi$ is $\varphi(\mathbf{x}_i)$. Multiplying $\mathcal{S}^{(k)}$ and $\Phi$ yields the **kernel sketch** $Z = \mathcal{S}^{(k)}\Phi$, where the $i$-th column of $Z^T$, denoted $\mathbf{z}_i$, is a linear combination of *all* node features in the kernel space. Crucially, these sketched embeddings $\{\mathbf{z}_i\}$ still preserve kernel relationships in expectation, i.e. for any $i, j$, $\mathbb{E}\big[\mathbf{z}_i^\top \mathbf{z}_j\big] = \kappa(\mathbf{x}_i, \mathbf{x}_j)$.

---

**Algorithm 1** Sketched Feature GNN

---

1: **Input** $\mathcal{G} = (V, E)$ ; $X \in \mathbb{R}^{N \times F}$; $k$;$L$
2: $\Phi = \mathcal{E}(X)$
3: $Z = \mathcal{S}^{(k)}(\Phi)$
4: Initialize: $\mathbf{h}_i^{(0)} = \mathbf{x}_i$ for all $i \in V$
5: **for** layer $\ell = 0$ to $L - 1$ **do**
6:     **for** each node $i \in V$ **do**
7:         $\tilde{\mathbf{h}}_i^{(\ell)} = [\mathbf{h}_i^{(\ell)} | \mathbf{z}_i]$
8:         $\mathbf{h}_i^{(\ell+1)} = f\left(\tilde{\mathbf{h}}_i^{(\ell)}, \{\tilde{\mathbf{h}}_j^{(\ell)} : j \in \mathcal{N}(i)\}\right)$
9: **return** $\mathbf{h}_i^{(L)}$

---

Interestingly, this sketched random feature matrix provides properties that, when injected into GNNs, *are surprisingly effective* in mitigating the aforementioned limitations of message-passing graph neural networks. In particular, these sketches:

- **are unique yet distance-sensitive.** Each sketch $\mathbf{z}_i$ is unique, breaking symmetries, while preserving distances in the node-feature space with high probability (Propositions 3.2 and 3.4). This property confers *universality* (enabling GNNs to distinguish non-isomorphic graphs) and counters *oversmoothing* by preventing embeddings from converging.
- **contain topology-agnostic cross-node information.** Since each $\mathbf{z}_i$ is formed by a linear combination of all node embeddings in the random feature space, it injects a *global* signal into the GNN (Proposition 3.3). This mitigates *oversquashing* by enabling immediate cross-node interactions, regardless of geodesic distance.
- **maintain equivariance in expectation.** With suitable random projections, the cross-node sketch preserves pairwise feature relationships under node permutation (Proposition 3.5).

In this work, we demonstrate analytically (Section 3) and empirically (Section 4) that these sketched random features provide the necessary global, distance-sensitive signals to overcome fundamental limitations of message-passing GNNs while maintaining computational efficiency. Unlike traditional uses of sketching for dimensionality reduction, our method leverages sketching as a mechanism for mixing global feature information across nodes, enabling efficient propagation of non-local signals without altering the graph topology. Experiments on real-world graph learning tasks demonstrate that SRF-augmented models often outperform baseline GNNs, offering a computationally efficient alternative to existing structural positional encodings. Moreover, by leveraging orthogonal information derived from node *features* rather than topology, our approach can be integrated with existing structural encoding methods to yield further performance improvements.

## 2 Background and Related Work

**Preliminaries.** We consider finite node-attributed graphs $\mathcal{G} = (V, E, X)$ with $|V| = N$ nodes, $|E|$ edges, and node feature matrix $X \in \mathbb{R}^{N \times F}$. Each column of $X^T$, $\mathbf{x}_i \in \mathbb{R}^F$, represents the features of node $i$. A *message-passing GNN (MPGNN)* [26] initializes each node representation with its features, i.e. for node $i$, $\mathbf{h}_i^0 = \mathbf{x}_i$, and iteratively updates each node's hidden representation $\mathbf{h}_i^{(\ell)}$ as

$$\mathbf{h}_i^{(\ell+1)} = f\left(\mathbf{h}_i^{(\ell)}, \{\mathbf{h}_j^{(\ell)} : j \in \mathcal{N}(i)\}\right), \tag{1}$$

where $\mathcal{N}(i)$ denotes the multiset of node i's neighbors and $f(\cdot)$ encapsulates learnable transformations (e.g., linear layers, MLPs) and aggregation mechanisms (e.g., sum, mean, attention). After $L$ layers, each node's hidden state $\mathbf{h}_i^{(L)}$ can be used for node-level tasks or pooled for graph-level tasks. Several popular MPGNN architectures [68, 40, 27, 66] can be considered special cases of this paradigm and differ only in specific choices of learnable functions and aggregations.

### 2.1 Known Limitations of GNNs

Despite the widespread adoption of MPGNNs, three fundamental limitations have attracted significant attention. We briefly describe each challenge below and provide a more comprehensive discussion and review of prior mitigation approaches for the limitations in Appendix A.1.

**Oversquashing.** As shown in Equation 1, MPGNNs propagate information through local neighborhoods. Consequently, when two nodes $i$ and $j$ have geodesic distance $d$, any signal from node $i$ requires at least $d$ message-passing layers to reach node $j$, and vice versa. During this process, the number of nodes in the effective receptive field can grow exponentially with depth, whereas the hidden dimension $|\mathbf{h}|$ typically remains fixed. This mismatch leads to *oversquashing* [2], where long-range information is compressed and loses influence on downstream tasks.

Alon and Yahav [2] demonstrate oversquashing empirically with their *Tree-NeighborsMatch* dataset, where a root node must predict a label based on leaf node signals traversing multiple tree levels. Despite the label being trivial to determine from direct leaf access, compression across the exponentially growing receptive field ($\mathcal{O}(2^L)$ at depth $L$) severely degrades performance as tree depth increases.

**Oversmoothing.** A commonly observed phenomenon in MPGNNs is the convergence of node embeddings $\mathbf{h}$ to an indistinguishable constant vector as network depth grows [61, 44, 56, 12]. In practice, this leads to GNNs being deployed with substantially fewer layers than their counterparts in other domains (e.g., deep convolutional neural networks [37]), despite depth often being critical for strong performance [19]. This effect is frequently attributed to the smoothing effect of repeated message-passing steps, which iteratively mix each node's features with those of its neighbors [44]. Although moderate smoothing can aid learning, it frequently converges rapidly to a subspace of trivial signals [56, 12], yielding nearly identical node representations and thus impeding metric performance. Following recent work [61, 12], we measure oversmoothing via a *Dirichlet energy* metric defined on node embeddings $H^{(\ell)} = \left[\mathbf{h}_1^{(\ell)}, \ldots, \mathbf{h}_N^{(\ell)}\right]^\top \in \mathbb{R}^{N \times F}$ after $\ell$ MPGNN layers (cf. Equation 1):

$$\mathcal{D}\left(H^{(\ell)}\right) = \frac{1}{N} \sum_{i \in \mathcal{V}} \sum_{j \in \mathcal{N}(i)} \left\|\mathbf{h}_i^{(\ell)} - \mathbf{h}_j^{(\ell)}\right\|_2^2, \tag{2}$$

A rapid decrease in $\mathcal{D}(H^{(\ell)})$ as $\ell$ grows indicates oversmoothing.

**Limited Expressiveness.** Ideally, GNNs should distinguish non-isomorphic graphs, as this is a pre-requisite for universal approximation of graph-invariant functions [53]. However, standard MPGNNs' expressiveness is upper-bounded by the 1-dimensional Weisfeiler–Lehman (1-WL) heuristic [67], which measures graph structure through iterative neighborhood aggregation. This expressivity constraint directly limits MPGNNs' ability to distinguish many non-isomorphic graphs that have distinct structural properties but identical neighborhood aggregation patterns [68]. In response, *k-order Weisfeiler–Lehman GNNs (k-WLGNNs)* were introduced [52, 50]. They operate on k-tuples of nodes and match the expressive power of the generalized k-WL heuristic, but typically require $\mathcal{O}(N^k)$ computation and memory. Although recent work has shown that this can be reduced in certain cases to $\mathcal{O}(N^2)$ and $\mathcal{O}(N^3)$ [4], these methods remain expensive for many applications.

To address this limitation efficiently, recent work proposes *positional* or *structural encodings* that augment node features with additional signals. For instance, purely *random* node features [18] render GNNs universal [62, 1], but often impair empirical performance [7]. Hence, recent efforts have shifted toward *graph-derived* features, seeking to make semantically meaningful encodings while retaining invariance to node permutations. Proposed approaches include spectral embeddings [23, 45, 48], subgraph-based representations [43, 9], and homomorphism counts [55, 34, 6]. However, designing an encoding that is simultaneously *unique*, *distance-sensitive*, and *equivariant* is difficult as it is closely tied to the *graph canonization* problem—known to be at least as hard as distinguishing graph isomorphism in general [5]. Recently, Pearl [36] proposes learning positional encodings with GNNs augmented with either standard basis vectors (B-Pearl) or random features (R-Pearl). This leads to better asymptotic complexity but can be expensive for small and medium graphs in practice.

In the following subsection, we introduce kernel methods and random feature approximations that serve as the foundation for our approach to address these limitations.

## 2.2 Kernels, Random Kernel Features, and Sketching

Consider a positive-definite function $\kappa : \mathbb{R}^F \times \mathbb{R}^F \to \mathbb{R}$. By the Moore–Aronszajn theorem [3], there exists a Hilbert space $\mathcal{H}$ and a *feature map* $\phi : \mathbb{R}^F \to \mathcal{H}$ such that

$$\kappa(\mathbf{x}, \mathbf{y}) = \langle \phi(\mathbf{x}), \phi(\mathbf{y}) \rangle_{\mathcal{H}} \quad \text{for all } \mathbf{x}, \mathbf{y} \in \mathbb{R}^d \tag{3}$$

In other words, any such function $\kappa$, known as a *kernel*, defines a lifting $\phi$ and inner product $\langle \cdot, \cdot \rangle$. This defines a geometry in $\mathcal{H}$ through the norm $\|\phi(\mathbf{x}) - \phi(\mathbf{y})\|_{\mathcal{H}}$, which in turn defines a notion of *distance* between $\mathbf{x}$ and $\mathbf{y}$ and thus $\kappa(\mathbf{x}, \mathbf{y})$ can be interpreted as similarity measure that we refer to as the *kernel similarity* between $\mathbf{x}$ and $\mathbf{y}$. For instance, for *linear kernel* $\kappa(\mathbf{x}, \mathbf{y}) = \mathbf{x}^\top \mathbf{y}$, the map $\phi$ is simply the identity, $\phi(\mathbf{x}) = \mathbf{x}$, and distance in $\mathcal{H}$ corresponds to standard Euclidean distance in $\mathbb{R}^d$.

Kernel methods [13, 30, 25] facilitate non-linear modeling by leveraging $\kappa$ to implicitly measure feature similarity in the space $\mathcal{H}$, thereby allowing otherwise linear models (e.g., SVMs [11]) to capture complex relationships without explicitly mapping data into that space. However, these methods typically require instantiation of *kernel matrix* $K_{i,j} = \kappa(\mathbf{x}_i, \mathbf{x}_j)$ for a dataset $\mathcal{X} = \{\mathbf{x}_1, \ldots, \mathbf{x}_N\} \subset \mathbb{R}^F$, which leads to $\mathcal{O}(N^2)$ memory complexity. Consequently, kernel methods often become impractical at large scales, motivating efficient approximations.

A powerful class of such approximations are *random kernel features*. These methods build on the famed *Johnson–Lindenstrauss lemma*, which proves the existence of a random linear map that approximately preserves pairwise distances in a dataset. Concretely, consider the same dataset $\mathcal{X} \subset \mathbb{R}^F$ from the last example. The JL lemma states that for any $\varepsilon \in (0, 1)$, there exists linear map $R \in \mathbb{R}^{D \times F}$ to dimension $D < F$ (with $D = \mathcal{O}(\frac{1}{\varepsilon^2} \log F)$) such that for all $i, j$

$$(1 - \varepsilon) \|\mathbf{x}_i - \mathbf{x}_j\|^2 \leq \|R\mathbf{x}_i - R\mathbf{x}_j\|^2 \leq (1 + \varepsilon) \|\mathbf{x}_i - \mathbf{x}_j\|^2 \tag{4}$$

A *Johnson–Lindenstrauss Transform (JLT)* implements such a linear map $R$ via random projection (e.g., a Gaussian matrix, which achieves the bound above whp). In the context of kernel approximations, one seeks a mapping $\varphi : \mathbb{R}^F \to \mathbb{R}^D$ that provides an unbiased estimate of the kernel:

$$\mathbb{E}\big[\varphi(\mathbf{x})^\top \varphi(\mathbf{y})\big] = \kappa(\mathbf{x}, \mathbf{y}). \tag{5}$$

In the simplest case of the *linear kernel*, $\kappa(\mathbf{x}, \mathbf{y}) = \mathbf{x}^\top \mathbf{y}$, we can directly take $\varphi_{\text{linear}}(\mathbf{x}) = R\mathbf{x}$, yielding a lower-dimensional estimation $\varphi_{\text{linear}}(\mathbf{x}) \in \mathbb{R}^D$. This provides an unbiased estimate of $\mathbf{x}^\top \mathbf{y}$ while reducing computational costs relative to an explicit $\mathcal{O}(N^2)$ kernel matrix.

More recent work has shown that *nonlinear* kernels can also be estimated via random transformations akin to the JLT [47]. In essence, one applies suitable nonlinearities to the randomly projected inputs, producing an unbiased approximation of a desired kernel (i.e. Equation 5). A prominent example is the use of *random Fourier features* [57], which enable efficient approximations of popular shift-invariant kernels such as the radial basis function (RBF) kernel $\phi_{\text{RBF}}$ and Laplacian kernel $\phi_{\mathcal{L}}$. Concretely, for the RBF kernel, one draws $D$ frequencies $\boldsymbol{\omega}_d$ for $d \in [D]$ from a Gaussian distribution, and applies a trigonometric mapping:

$$\varphi_{\text{RBF}}(\mathbf{x}) \;=\; \sqrt{\frac{2}{D}} \left[ \cos(\boldsymbol{\omega}_1^\top \mathbf{x} + b_1), \, \ldots, \, \cos(\boldsymbol{\omega}_D^\top \mathbf{x} + b_D) \right]^\top, \tag{6}$$

where $b_k \in [0, 2\pi)$ are sampled uniformly at random. Similarly, $\phi_{\mathcal{L}}$ is approximated by sampling $\boldsymbol{\omega}$ from a Cauchy distribution and applying the same transformation.

In the next section, we apply these ideas in two distinct ways. First, we apply kernel embedding approximations (e.g., Equation 6) to estimate the feature map $\phi$. Second, we apply JLT projections to the resulting kernel embeddings and show that doing so leads to desirable qualities when augmenting MPGNNs. To avoid confusion, we distinguish these two transformations clearly: the first set are **kernel** transformations of node features and the second are **Sketched Random Features (SRF)**.

## 3 Sketched Random Features

### 3.1 Defining SRF

Building on the ideas discussed in Section 1-2, our goal is to construct *Sketched Random Features* that (1) are distance-sensitive (2) unique, (3) encode signals across all nodes, and (4) remain permutation-equivariant under node relabelings.

**Embedding Operator $\mathcal{E}$.** Let $X \in \mathbb{R}^{N \times F}$ be the raw feature matrix, with each column of $X^T$, denoted $\mathbf{x}_i$, corresponding to node $i$. We define an embedding operator $\mathcal{E} : \mathbb{R}^{N \times F} \to \mathbb{R}^{N \times D}$ that applies function $\varphi : \mathbb{R}^F \to \mathbb{R}^D$ to each column of $X^T$ independently. That is, $\mathcal{E}(X^T)_{:,i} = \varphi(\mathbf{x}_i)$.

In this work, we focus on $\varphi$ functions that map features to random embeddings whose inner products yield unbiased estimates (Equation 5) of a kernel $\kappa$ (Equation 3). Examples are $\varphi_{\text{linear}}$, $\varphi_{\mathcal{L}}$, and $\varphi_{\text{RBF}}$, as discussed in Section 2.2, yielding embedding operators $\mathcal{E}_{\text{linear}}$, $\mathcal{E}_{\mathcal{L}}$, and $\mathcal{E}_{\text{RBF}}$, respectively. Hence, we define the *kernel feature matrix* as

$$\Phi \;=\; \mathcal{E}(X) \;\in\; \mathbb{R}^{N \times D}.$$

**Sketch Operator $\mathcal{S}$.** We next introduce a sketch operator $\mathcal{S} : \mathbb{R}^{N \times D} \to \mathbb{R}^{N \times D}$. This random projection matrix implements a Johnson Lindenstrauss Transform (JLT) (Section 2.2) which provides approximate preservation of pairwise distances with high probability (see Equation 4). In this work, we focus on the *additive Gaussian* (AG) sketch, which we define as:

$$\mathcal{S}_{\text{AG}}(\Phi) \;=\; \left( I + \frac{1}{\sqrt{N}} G \right) \Phi, \tag{7}$$

where $I \in \mathbb{R}^{N \times N}$ is the identity matrix, and $G \in \mathbb{R}^{N \times N}$ has i.i.d. entries $G_{ij} \sim \mathcal{N}(0, 1)$. Multiplying $\Phi$ by the sketch forms random linear combinations of all rows in $\Phi$.

Unlike typical JLT applications, we *do not* reduce dimension $N$. Instead, we employ this random projection because it possesses properties (Propositions 3.1–3.5) that we show in Section 3.2 can mitigate the standard limitations of message-passing GNNs. Importantly, the dimension $D$ is determined by the embedding operator $\mathcal{E}$ rather than $\mathcal{S}$.

**Multi-Projection Sketching.** $\mathcal{S}_{\text{AG}}$ introduces a one dimensional random projection of each column of $\Phi$. We can generalize this to multiple dimensions via concatenation. Specifically, we introduce the $k$-order sketch operator $\mathcal{S}_{\text{AG}}^{(k)} : \mathbb{R}^{N \times D} \to \mathbb{R}^{N \times kD}$:

$$\mathcal{S}_{\text{AG}}^{(k)}(\Phi) \;=\; \left[ \left( I + \tfrac{1}{\sqrt{N}} G^{(1)} \right) \Phi \,\middle|\middle|\, \left( I + \tfrac{1}{\sqrt{N}} G^{(2)} \right) \Phi \,\middle|\middle|\, \ldots \,\middle|\middle|\, \left( I + \tfrac{1}{\sqrt{N}} G^{(k)} \right) \Phi \right]$$

where each $G^{(k)} \in \mathbb{R}^{N \times N}$ is drawn independently with i.i.d. $\mathcal{N}(0, 1)$ entries. This operation concatenates $k$ independent AG sketches, enabling k-dimensional estimates of each feature.

**Sketched Random Features (SRF).** Bringing these components together, we define the *kernel sketch* matrix $Z$ as

$$Z = \mathcal{S}^{(k)}(\mathcal{E}(X)) \in \mathbb{R}^{N \times (kD)}, \tag{8}$$

where the new feature embedding of node $i$ is given by $\mathbf{z}_i \triangleq Z_{i,:}$.[1] In our experiments (Section 4), we primarily analyze the (multi-) projection AG sketch $\mathcal{S}_{\text{AG}}^{(k)}$ as introduced above, but also consider the trivial identity sketch $\mathcal{S}_{\text{id}} = I$ as an ablation study. For embedding operator $\mathcal{E}$, we consider the operators based on the random kernel features discussed in Section 2.2: $\mathcal{E}_{\text{linear}}, \mathcal{E}_{\mathcal{L}},$ and $\mathcal{E}_{\text{RBF}}$.

## 3.2 Enhancing MPGNNs with Node Feature Sketches

We next turn to incorporate SRF into MPGNNs. To do so, we augment the node hidden states $\mathbf{h}_i^{(\ell)}$ at each layer with the corresponding sketched embedding $\mathbf{z}_i$ via concatination. Let $\widetilde{\mathbf{h}}_i^{(\ell)} = [\mathbf{h}_i^{(\ell)} | \mathbf{z}_i]$ denote this augmented representation, where $[\cdot | \cdot]$ represents vector concatenation along the feature dimension. The updated MPGNN formulation becomes (cf. Equation 1):

$$\mathbf{h}_i^{(\ell+1)} = f(\widetilde{\mathbf{h}}_i^{(\ell)}, \{\widetilde{\mathbf{h}}_j^{(\ell)} : j \in \mathcal{N}(i)\}). \tag{9}$$

By injecting $\mathbf{z}_i$ at each layer, the model has access to both local node features and the information encoded by SRF. This strategy addresses common MPGNN weaknesses discussed in Section 2, as detailed in the proceeding subsection. Algorithm 1 summarizes the SRF-enhanced GNN procedure.

## 3.3 Properties of Sketched Random Features

We now establish key theoretical properties[2] of SRF and demonstrate how they address core limitations of message-passing GNNs. The following propositions characterize the fundamental mathematical properties of our approach.

**Proposition 3.1** (Unbiased Cross-Terms in the Kernel Matrix). *SRF provides an unbiased estimation of cross-terms in the Kernel matrix. Specifically, for any distinct nodes $i$ and $j$, the cross-term of the Gram matrix is unbiased: $\mathbb{E}_{X,G}[\mathbf{z}_i^\top \mathbf{z}_j] = \kappa(\mathbf{x}_i, \mathbf{x}_j)$.*

*Proof.* From Equation 7, the $(i, j)$-th inner product is

$$\mathbf{z}_i^\top \mathbf{z}_j = \sum_{p,q=1}^{N} (I_{ip} + \tfrac{1}{\sqrt{N}} G_{ip})(I_{jq} + \tfrac{1}{\sqrt{N}} G_{jq}) \, \varphi(\mathbf{x}_p)^\top \varphi(\mathbf{x}_q).$$

We note that for $i \neq j$, $\mathbb{E}[G_{ip}] = 0$ and $\mathbb{E}[G_{ip} G_{jq}] = \delta_{ij} \delta_{pq} = 0$. Thus, by Equation 5, we have

$$\mathbb{E}_{X,G}[\mathbf{z}_i^\top \mathbf{z}_j] = \sum_{p,q=1}^{N} I_{ip} I_{jq} \, \mathbb{E}[\varphi(\mathbf{x}_p)^\top \varphi(\mathbf{x}_q)] = \mathbb{E}[\varphi(\mathbf{x}_i)^\top \varphi(\mathbf{x}_j)] = \kappa(\mathbf{x}_i, \mathbf{x}_j).$$

$\square$

**Proposition 3.2** (Kernel Distance Sensitivity). *With high probability, there exists a positive $c \sim \mathcal{O}(N^{-1/2})$ such that for any nodes $i, j$:*

$$(1 - c)\|\varphi(\mathbf{x}_i) - \varphi(\mathbf{x}_j)\|_2 \leq \|\mathbf{z}_i - \mathbf{z}_j\|_2 \leq (1 + c)\|\varphi(\mathbf{x}_i) - \varphi(\mathbf{x}_j)\|_2$$

*Proof.* By construction, $\mathbf{z}_i - \mathbf{z}_j = \left(I + \tfrac{1}{\sqrt{N}} G\right) \left(\varphi(\mathbf{x}_i) - \varphi(\mathbf{x}_j)\right)$ since each row $\mathbf{z}_i$ corresponds to the $i$-th row of $\left(I + \tfrac{1}{\sqrt{N}} G\right) \mathcal{E}(X)$. If $\varphi(\mathbf{x}_i) = \varphi(\mathbf{x}_j)$, then $\|\mathbf{z}_i - \mathbf{z}_j\|_2 = 0$ and the claim holds trivially. Thus, we assume $\varphi(\mathbf{x}_i) \neq \varphi(\mathbf{x}_j)$ and define $\mathbf{v} = \varphi(\mathbf{x}_i) - \varphi(\mathbf{x}_j)$. Applying the triangle inequality:

$$\left| \|\mathbf{v}\|_2 - \tfrac{1}{\sqrt{N}} \|G\mathbf{v}\|_2 \right| \leq \left\| \mathbf{v} + \tfrac{1}{\sqrt{N}} G\mathbf{v} \right\|_2 \leq \|\mathbf{v}\|_2 + \tfrac{1}{\sqrt{N}} \|G\mathbf{v}\|_2$$

---

[1] In the featureless limit where all $\mathbf{x}_i$ are identical, SRF degenerates to random node individualization [62, 1], preserving expressive power through randomization.

[2] For notational clarity, we present proofs under base case $\mathcal{S}_{\text{AG}}^{(1)}$ and as the extension to $\mathcal{S}_{\text{AG}}^{(k)}$ is straightforward.

Next, the JL lemma (Equation 4) guarantees that, given $\varepsilon \in [0, 1]$, with high probability over the choice of the random Gaussian matrix $G$[3]:

$$\sqrt{1-\varepsilon}\,\|\mathbf{v}\|_2 \;\leq\; \tfrac{1}{\sqrt{N}}\|G\,\mathbf{v}\|_2 \;\leq\; \sqrt{1+\varepsilon}\,\|\mathbf{v}\|_2.$$

Substituting these bounds back into the triangle inequality shows

$$\left(1 - \sqrt{\frac{1+\varepsilon}{N}}\right) \leq \frac{\|\mathbf{z}_i - \mathbf{z}_j\|_2}{\|\mathbf{v}\|_2} \leq \left(1 + \sqrt{\frac{1+\varepsilon}{N}}\right).$$

Thus, letting $c = \sqrt{(1+\varepsilon)/N}$ completes the proof. $\qquad\square$

**Proposition 3.3** (Cross-Node Information). *For any node $i$, the sketched embedding $\mathbf{z}_i$ contains a linear combination of the embeddings of* all *nodes in the graph.*

*Remark.* Proposition 3.3 ensures that each SRF embedding encodes cross-node information due to the row sketch construction.

**Proposition 3.4** (Almost Sure Uniqueness). $\{\mathbf{z}_i\}$ *are unique with probability 1.*

*Proof.* Without loss of generality, consider the case where the node features are identical, e.g. $\mathbf{x}_i = \mathbf{1}$. Then $\varphi(\mathbf{x}_i) = \varphi(\mathbf{1})$ for all $i$, and from the definition of SRF with $\mathcal{S}_{AG}^{(1)}$ we have for row $i$ of $Z$:

$$\mathbf{z}_i = \varphi(\mathbf{1}) + \frac{1}{\sqrt{N}} \sum_{j=1}^{N} G_{ij}\varphi(\mathbf{1})$$

Since $\{\sum_{j=1}^{N} G_{ij}\}$ are independent Gaussian random variables, the coefficients multiplying $\varphi(\mathbf{1})$ are almost surely unique. Thus, $\mathbf{z}_i \neq \mathbf{z}_j$ for $i \neq j$ with probability 1. In the general case where node features differ, $\varphi(\mathbf{x}_i)$ and $\varphi(\mathbf{x}_j)$ may also differ, providing additional distinctness. $\qquad\square$

**Proposition 3.5** (Permutation Equivariance in Expectation). *Let $\pi$ be a fixed permutation of node indices, with corresponding permutation matrix $P_\pi$. Let the function $f \triangleq \mathcal{S}_{AG} \circ \mathcal{E}$ be the SRF operation (Equation 8). Then $f$ is permutation equivariant in expectation: $\mathbb{E}_{X,G}[f(P_\pi X)] = \mathbb{E}_{X,G}[P_\pi f(X)]$.*

*Proof.* By definition, $f(X) = \left(I + \frac{1}{\sqrt{N}}G\right)\Phi(X)$ and $\mathbb{E}[G] = 0$. Then

$$\mathbb{E}[f(P_\pi X)] = \mathbb{E}\left[\left(I + \tfrac{1}{\sqrt{N}}G\right)P_\pi \Phi(X)\right] = P_\pi \Phi(X),$$

and similarly

$$\mathbb{E}[P_\pi f(X)] = \mathbb{E}\left[P_\pi \left(I + \tfrac{1}{\sqrt{N}}G\right)\Phi(X)\right] = P_\pi \Phi(X).$$

$\qquad\square$

**SRF Mitigates Oversquashing.** SRF mitigates oversquashing by encoding topology-agnostic cross-node information in each sketched embedding $\mathbf{z}_i$, as shown in Proposition 3.3. While this encoding into $\mathbb{R}^D$ (where typically $D \ll N$) is not lossless, it enables direct information flow between all node pairs independent of their geodesic distance. This circumvents the exponential information bottleneck inherent in multi-hop message passing [2]. Our empirical analysis in Section 4.1 demonstrates that SRF achieves linear rather than exponential information loss as $N$ increases.

**SRF Alleviates Oversmoothing.** SRF addresses oversmoothing by introducing node-specific, distance-sensitive embeddings that preserve variance across nodes (Propositions 3.1 and 3.2, respectively). The inclusion of unique $\mathbf{z}_i$ vectors (Proposition 3.4) repeatedly injected into message passing (Equation 9) ensures $\widetilde{\mathbf{h}}_i^{(\ell)} = [\mathbf{h}_i^{(\ell)}|\mathbf{z}_i]$ remain distinct as $\ell$ grows. Our empirical results (Section 4.1) show this approach maintains higher representation diversity even in base features $\mathbf{h}_i^{(\ell)}$, preventing representational collapse with depth.

---

[3]Note that while the version of the JL lemma discussed in Section 2.2 (Equation 4) provides bounds for squared norms, taking square roots is valid since all terms are positive.

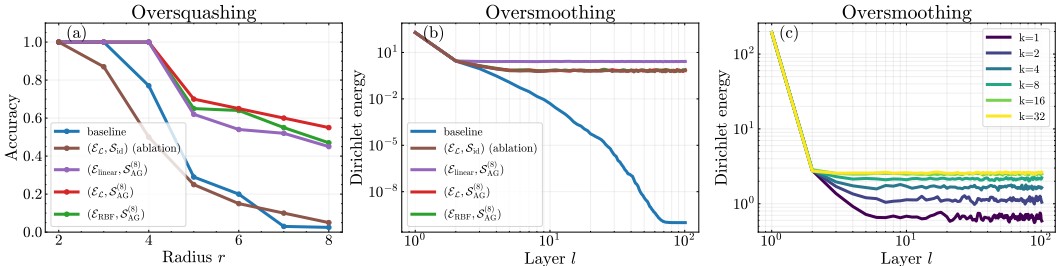

Figure 1: Analysis of oversquashing and oversmoothing across model variants. (a) Accuracy vs. graph radius showing the effect of oversquashing across methods. (b) Dirichlet energy across layers for different methods. (c) Dirichlet energy across for $\mathcal{E}_{\mathcal{L}}$ with varying $k$. Lower Dirichlet energy indicates more oversmoothing.

**Universality of SRF in MPGNNs.** SRF provides a principled way to achieve universality in MPGNNs. As discussed in Section 2, purely random features guarantee universality, but often fail to improve metric performance in practice. In contrast, structural encodings provide semantically meaningful and equivariant signals which lead to improved empirical performance, but often fail to provide an encoding that is unique, distance sensitive and equivariant, due to the inherent difficulty of the graph canonization problem. SRF strikes a balance between these approaches: it is unique (Proposition 3.4), ensuring universality, while preserving distance in the node feature kernel space whp (Proposition 3.2). Additionally, it is permutation equivariant in expectation (Proposition 3.5).

**Memory and Runtime Complexity.** While the JLT is traditionally implemented with dense random projection matrices, memory and runtime efficiency can be improved by using structured random matrices (SRMs) [15] as detailed in Appendix A.2. SRMs lower storage requirements to $\mathcal{O}(N)$ and enable matrix-vector multiplication in $\mathcal{O}(N \log N)$. Notably, these matrices preserve the theoretical guarantees of Section 3. This yields favorable asymptotic complexity compared to many positional encoding methods. Detailed complexity analysis and comparisons are in Appendix A.2.

## 4 Experiments

We empirically evaluate SRF-enhanced GNNs [4], first validating that SRF addresses key MPGNN limitations Section 4.1, then demonstrating performance on real-world benchmarks (Section 4.2). We use GIN [68] for unattributed-edge graphs and GINE [31] for edge-featured graphs, with additional architectures tested in Appendix B to verify architecture-agnostic benefits.

We evaluate embedding operators $\mathcal{E}_{\mathrm{linear}}$, $\mathcal{E}_{\mathcal{L}}$, and $\mathcal{E}_{\mathrm{RBF}}$ with additive Gaussian sketch $\mathcal{S}_{\mathrm{AG}}^{(k)}$, denoting configurations as $\left(\mathcal{E}, \mathcal{S}_{\mathrm{AG}}^{(k)}\right)$. To isolate sketching effects from kernel features alone, we ablate with identity operator $\mathcal{S}_{\mathrm{id}} = I$, comparing $\left(\mathcal{E}_{\mathcal{L}}, \mathcal{S}_{\mathrm{id}}\right)$ against $\left(\mathcal{E}_{\mathcal{L}}, \mathcal{S}_{\mathrm{AG}}^{(k)}\right)$. Full dataset descriptions, baseline details, hyperparameter search procedures, and other experimental details are provided in Appendix C. We include additional experiments in the Appendix: validation of SRF benefits across diverse GNN architectures (Appendix B) and analysis of SRF hyperparameters (embedding dimension $D$ and projection count $k$) on performance (Appendix D).

### 4.1 Synthetic Learning Tasks

**Oversquashing.** We evaluate SRF's ability to mitigate oversquashing using the *Tree-NeighborsMatch* synthetic benchmark [2] discussed in Section 2. Recall that as radius $r$ increases, exponentially more information must be aggregated by the message passing algorithm, making the task progressively harder for standard MPGNNs. Results in Figure 1 show that while baseline GIN and ablation $\left(\mathcal{E}_{\mathcal{L}}, \mathcal{S}_{\mathrm{id}}\right)$ suffer severe performance degradation beyond $r = 4$, SRF-enhanced models maintain higher accuracy at large radii, regardless of choice of $\mathcal{E}$. Notably, all SRF variants achieve

---

[4]Our source code is available at https://github.com/ryienh/sketched-random-features.

Table 1: Performance on synthetic expressiveness benchmarks. All results averaged over 5 runs. Standard deviations ($\leq 0.002$ across all experiments) omitted for clarity.

| Dataset | Baseline | Ablation | Ours ($\mathcal{S}_{\text{AG}}^{(1)}$) | | | Ours ($\mathcal{S}_{\text{AG}}^{(8)}$) | | |
|---|---|---|---|---|---|---|---|---|
| | None | $(\mathcal{E}_{\mathcal{L}}, \mathcal{S}_{\text{id}})$ | $\mathcal{E}_{\text{linear}}$ | $\mathcal{E}_{\mathcal{L}}$ | $\mathcal{E}_{\text{RBF}}$ | $\mathcal{E}_{\text{linear}}$ | $\mathcal{E}_{\mathcal{L}}$ | $\mathcal{E}_{\text{RBF}}$ |
| CSL (Acc ↑) | 0.100 | 0.100 | 1.000 | 1.000 | 1.000 | 1.000 | 1.000 | 1.000 |
| EXP (Acc ↑) | 0.518 | 0.520 | 1.000 | 1.000 | 1.000 | 1.000 | 1.000 | 1.000 |

perfect accuracy up to $r = 4$ and demonstrate more graceful performance decay thereafter, with accuracy remaining above 40% even at $r = 8$. This improved performance at large radii empirically validates our theoretical analysis that SRF enables comparatively more effective long-range communication in MPGNNs.

**Oversmoothing.** To assess SRF against oversmoothing, we follow Rusch et al. [61]'s protocol using Cora [51] with randomized features ($X_{jk} \sim \mathcal{N}(0, 1)$) and measuring Dirichlet energy (Equation 2). Figure 1(b) shows baseline GNNs exhibit near-exponential energy decay while SRF variants maintain substantially higher energy. While feature injection alone helps prevent decay (ablation study), Figure 1(c) shows that increasing projections ($k$) further preserves node distinctness in deeper layers.

**Expressiveness.** We evaluate the universal approximation capabilities of SRF-enhanced MPGNNs using two graph isomorphism discrimination benchmarks: CSL [54] and EXP [1], containing graphs indistinguishable by 1-WL and 2-WL tests respectively. Table 1 shows SRF-enhanced models achieving perfect discrimination on both datasets, while baselines and ablations (without sketching) perform no better than random chance.

## 4.2 Real-world Graph Learning Tasks

We evaluate SRF-enhanced GNNs on real-world datasets, benchmarking against existing positional encoding approaches (Section 2.1). Our results show SRF-enhanced GNNs *outperform many positional encoding approaches* (Table 2, left) while offering *substantial efficiency gains*: about 3 times faster runtime and about two orders of magnitude less memory than PEARL on evaluated datasets (Figure 2). They show *robustness in out-of-distribution tasks* where other approaches falter, remaining competitive with state-of-the-art approaches (Table 2, right). Finally, they can be *combined with positional encodings for cumulative improvements* due to their complementary nature (Table 3).

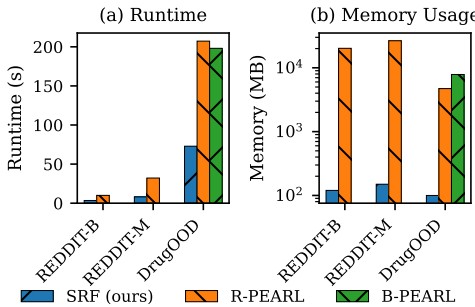

(a) Runtime    (b) Memory Usage

Figure 2: Training efficiency comparison of SRF (ours), R-PEARL, and B-PEARL. (a) Runtime in seconds. (b) Memory usage in MB (log scale).

**Baselines.** We consider several positional encoding baselines discussed in Section 2.1 and Appendix A: random node features (rand id) [1, 62], SignNet and BasisNet [45], efficient approximations of SignNet (SignNet-8S, SignNet-8L) [36], SPE [32] with its efficient variant SPE-8S, and Pearl (R-PEARL, B-PEARL) [36]. For some experiments, we include graph transformers and attention-enhanced GNNs: GPS [58], SAN [41], SUN [24], GNN-AK [71], Graph ViT [29].

**Social Network Classification.** We evaluate on REDDIT-B and REDDIT-M datasets [69], which represent online discussion threads where nodes are users and edges indicate comment interactions. These datasets require models to classify discussion graphs into their respective subreddits (online communities). As shown in Table 2, SRF-augmented GNNs consistently outperform baseline methods. Notably, even our simplest variant ($\mathcal{E}_{\text{linear}}$) demonstrates substantial improvements over prior state-of-the-art methods, with $\mathcal{E}_{\text{RBF}}$ providing the strongest performance. The ablation confirms these gains derive from sketching rather than merely adding kernel features.

Table 2: Performance comparison of SRF enhanced GNNs and several positional encoding baselines on Reddit (% accuracy) and DrugOOD (% AUC) datasets. Results for baselines from Kanatsoulis et al. [36]. Values missing from the literature are denoted by "—". OOM indicates out of memory. First, second, and third best results are highlighted in blue, green, and orange, respectively.

| | Reddit (Acc ↑) | | DrugOOD (AUC ↑) | | |
|---|---|---|---|---|---|
| | REDDIT-B | REDDIT-M | Assay | Scaffold | Size |
| *Baselines:* | | | | | |
| GINE (No PE) | $91.8 \pm 1.0$ | $56.9 \pm 2.0$ | $71.68 \pm 1.10$ | $68.00 \pm 0.60$ | $66.04 \pm 0.70$ |
| GINE + rand id | $91.8 \pm 1.6$ | $57.0 \pm 2.1$ | — | — | — |
| SignNet | OOM | OOM | $72.27 \pm 0.97$ | $66.43 \pm 1.06$ | $64.03 \pm 0.70$ |
| SignNet-8S | $92.4 \pm 1.1$ | $57.8 \pm 0.8$ | — | — | — |
| SignNet-8L | $79.5 \pm 12.3$ | $41.4 \pm 2.7$ | — | — | — |
| BasisNet | — | — | $71.66 \pm 0.05$ | $66.32 \pm 5.68$ | $60.79 \pm 3.19$ |
| SPE | — | — | $72.53 \pm 0.66$ | $69.64 \pm 0.49$ | $66.02 \pm 0.49$ |
| SPE-8S | — | — | $71.72 \pm 0.71$ | $68.72 \pm 0.63$ | $65.74 \pm 2.20$ |
| R-PEARL | $93.0 \pm 1.3$ | $59.4 \pm 1.0$ | $72.24 \pm 0.30$ | $69.20 \pm 1.00$ | $65.89 \pm 1.30$ |
| B-PEARL | — | — | $71.22 \pm 0.42$ | $69.51 \pm 0.62$ | $66.58 \pm 0.67$ |
| *Ablation:* | | | | | |
| $(\mathcal{E}_{\mathcal{L}}, \mathcal{S}_{\text{id}})$ | $92.56 \pm 0.58$ | $58.32 \pm 0.47$ | $71.89 \pm 0.37$ | $65.34 \pm 0.26$ | $63.29 \pm 0.63$ |
| *Ours:* $(\mathcal{S}_{\text{AG}}^{(8)})$ | | | | | |
| $\mathcal{E}_{\text{linear}}$ | $94.06 \pm 0.39$ | $60.18 \pm 0.39$ | $72.29 \pm 0.30$ | $68.79 \pm 0.64$ | $66.45 \pm 0.24$ |
| $\mathcal{E}_{\mathcal{L}}$ | $94.00 \pm 0.35$ | $60.33 \pm 0.37$ | $72.51 \pm 0.69$ | $69.43 \pm 0.90$ | $67.23 \pm 0.54$ |
| $\mathcal{E}_{\text{RBF}}$ | $94.13 \pm 0.69$ | $60.53 \pm 0.29$ | $72.63 \pm 0.41$ | $69.60 \pm 0.48$ | $66.67 \pm 0.35$ |

**Molecular Graph Out-of-Distribution (OOD) Generalization.** We evaluate OOD generalization on DrugOOD [33], which tests molecular property prediction across three domain shifts (Assay, Scaffold, Size). As Kanatsoulis et al. [36] demonstrate, these OOD tasks are particularly challenging for positional encoding approaches, with many advanced methods actually degrading performance compared to GNN backbones alone. Conversely, Table 2 demonstrates that our *feature-based* augmentation approach effectively overcomes this limitation, with all SRF variants consistently outperforming the baseline GINE model across all splits. Notably, our approach achieves state-of-the-art performance on the Assay and Size shifts, and remains competitive on the Scaffold split.

**Long-Range Interactions in Peptide Structure Prediction.** To assess whether SRF complements structural approaches, we evaluate on Peptides-struct [22], a benchmark with long-range dependencies where graph transformers typically excel. Table 3 shows that combining PEARL positional encodings and SRF augmentation $(\mathcal{E}_{\text{RBF}}, \mathcal{S}_{\text{AG}}^{(8)})$ closes the performance gap between graph transformers and GNNs, verifying SRF provides complementary benefits to positional encodings.

Table 3: Performance comparison for peptide-struct (MAE, lower is better). Baseline results reported by Kanatsoulis et al. [36]. Error bars presented in Appendix C but do not exceed $\pm 0.004$.

| Baselines | | | | | | | Ours | |
|---|---|---|---|---|---|---|---|---|
| R-PEARL | B-PEARL | GPS | SAN+ RWSE | GNN-AK+ | SUN | Graph ViT | R-PEARL +SRF | B-PEARL +SRF |
| 0.247 | 0.248 | 0.252 | 0.255 | 0.274 | 0.250 | 0.245 | 0.245 | 0.243 |

## 5 Conclusion

We have presented an unconventional application of the Johnson-Lindenstrauss transform for enhancing message-passing graph neural networks. Our theoretical analysis and empirical results demonstrate that such a strategy overcomes well-known shortcomings of MPGNNs with minimal computational overhead, making it a practical enhancement for existing architectures. A promising direction for future work is to investigate whether analogous sketching techniques generalize to topology-aware embeddings (e.g., graph random kernel features [16]) or structural encodings.

## Acknowledgments and Disclosure of Funding

This research used resources of the Argonne Leadership Computing Facility, a U.S. Department of Energy (DOE) Office of Science user facility at Argonne National Laboratory and is based on research supported by the U.S. DOE Office of Science-Advanced Scientific Computing Research Program, under Contract No. DE-AC02-06CH11357. Additional funding support comes from the National Science Foundation (CCF-2119184 CNS-2313190 CCF-1822949 CNS-1956180). RW gratefully acknowledges the support of NSF DMS-2023109, DOE DE-SC0022232, the NSF-Simons National Institute for Theory and Mathematics in Biology (NITMB) through NSF (DMS-2235451) and Simons Foundation (MP-TMPS-00005320), and the Margot and Tom Pritzker Foundation.

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

# A  SRF Background and Complexity Analysis

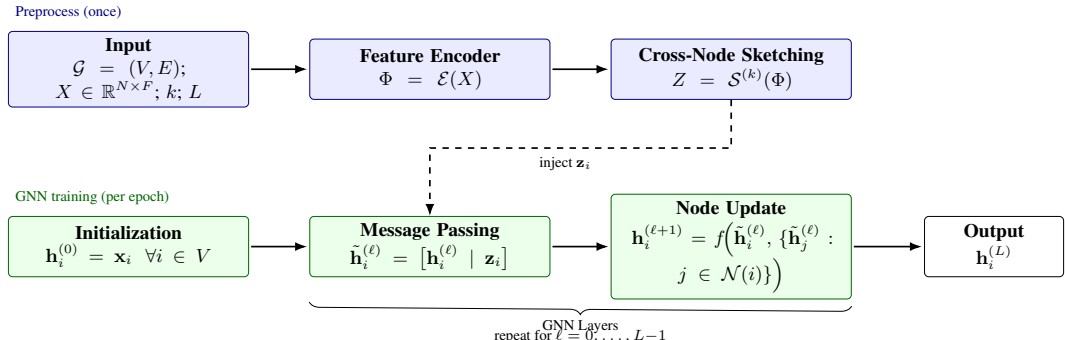

Figure 3: Block diagram visualizing the SRF method defined in Algorithm 1. SRF is computed once (top, blue) and then concatenated to node states at every GNN layer during training (bottom, green).

This section expands on the limitations of MPGNNs discussed in Section 2.1, reviews prior mitigation strategies, and compares the computational complexity of SRF with recent baselines. Figure 3 overviews the SRF preprocessing strategy defined in Algorithm 1 and described in detail in Section 3.

## A.1  Additional Discussion on Prior Work

**Oversquashing.**   Theoretical work formalizes oversquashing by analyzing the partial derivative of the hidden states of nodes $i$ and $j$ at depth $l$, i.e. $\left|\partial \mathbf{h}_i^{(l)} / \partial \mathbf{h}_j^0\right|$, with Topping et al. [64] showing it is bounded by $c \cdot A^d$ for constant $c$, graph adjacency matrix $A$, and geodesic distance $d$, thereby implying that node $\mathbf{h}_i$ and $\mathbf{h}_j$ become exponentially less sensitive to each other as $d$ grows. Furthermore, Di Giovanni et al. [20] validate the intuition posed in Alon and Yahav [2], proving that increasing hidden dimension $|\mathbf{h}|$ can alleviate oversquashing, whereas greater depth $|L|$ alone fails to resolve it.

Recent work seeks to mitigate oversquashing in MPGNNs by defining a *rewiring* $\mathcal{Q} : \mathbb{R}^{n \times n} \to \mathbb{R}^{n \times n}$ that operates on the adjacency matrix $A$ to yield $\mathcal{Q}(A)$, hence producing a rewired graph $\mathcal{G}'$. This process reduces either the graph diameter or the Cheeger constant (which quantifies the connectivity of a graph relative to its cut size) [17][5]. Thus, these strategies alleviate oversquashing by altering topology to remove bottlenecks. However, this leads to an altered and usually denser graph $\mathcal{G}'$, undermining the original topological bias of message passing and increasing computational overhead.

**Oversmoothing.**   Common ways to mitigate oversmoothing include strategies based on regularization [28, 72, 60] or skip connections [42, 14]. Nevertheless, striking a balance between preventing oversmoothing and preserving the expressive power of deeper GNNs remains challenging. Indeed, recent evidence suggests that merely suppressing oversmoothing without enhancing a model's capacity may still limit its ultimate performance [61]. We refer interested readers to the recent survey by Rusch et al. [61] for a more detailed discussion.

**Limited Expressiveness.**   A pedagogical example of the shortcomings of existing positional encodings described in Section 2 is the use of Laplacian eigenvectors (e.g. [23]), which can produce multiple valid solutions depending on the chosen sign or basis, and may fail to distinguish co-spectral but non-isomorphic graphs. This limitation motivates a body of existing encoding approaches such as learned [50] and heuristic-based [21] eigenvector canonization. However, as described in the main manuscript, designing encodings based on topology that are unique, distance sensitive, and equivariant in polynomial time may be impossible due to its relatedness to graph canonization [5]. We refer interested readers to the recent survey by Morris et al. [53] for a more comprehensive overview of positional encodings for GNNs.

As described in the main manuscript, recent work presented by Kanatsoulis et al. [36] instead aim to *learn* positional encodings using a message passing module that is trained end-to-end with the main

---

[5]We credit Di Giovanni et al. [20] for unifying these prior approaches and offering a comprehensive overview.

("backbone") GNN. The PEARL methodology begins by anonymizing the input graph by stripping away all node and edge attributes. For each node in the graph, the framework generates M random (R-PEARL) or basis (B-PEARL) attributes. These generated samples are then individually processed through a GNN and its outputs are combined using a pooling function $\rho$ to create (equivariant) positional encodings. In the final stage, the framework processes the graph structure alongside the generated PEs and any existing node or graph attributes, using either a GNN or a Graph Transformer for the final analysis. Thus, PEARL's positional encodings enjoy better asymptotic complexity than many classical (e.g. spectral-based) positional encoding methods, including a linear, rather than cubic cost, in the number of nodes in the input graph. However, unlike these classical methods, PEARL pays this $\mathcal{O}(n)$ cost for each graph at every epoch of training.

## A.2    Complexity Analysis

**SRF Complexity and SRMs.** We analyze the runtime complexity of computing Sketched Random Features (SRF) when using structured random matrices (SRMs) as the sketching operator, as introduced in Section 3 of the main manuscript. SRMs, introduced by Choromanski et al. [15], are designed to accelerate Johnson–Lindenstrauss-style projections while preserving statistical properties of dense Gaussian matrices. Specifically, SRMs such as SD-product matrices (e.g., Hadamard–Rademacher constructions) admit fast matrix-vector multiplication in time $\mathcal{O}(N \log N)$, rather than the $\mathcal{O}(N^2)$ cost incurred by unstructured dense matrices. When applied to a full matrix $\Phi \in \mathbb{R}^{N \times D}$ consisting of $D$ feature vectors, the corresponding matrix-matrix multiplication with a SRM costs $\mathcal{O}(N^2 \log N)$.

Our SRF construction proceeds in two stages. Given an input feature matrix $X \in \mathbb{R}^{N \times F}$, we first compute kernel embeddings by applying a random feature map $\phi : \mathbb{R}^F \to \mathbb{R}^D$ (e.g., random Fourier features [57]) to each row of $X$, yielding the matrix $\Phi = E(X) \in \mathbb{R}^{N \times D}$. This step has complexity $\mathcal{O}(NFD)$ under the assumption that $\phi$ involves dense linear projections.

In the second stage, we apply a sketching operator $S^{(k)}$ consisting of $k$ independent SRMs to $\Phi$, producing the final sketched matrix $Z = S^{(k)}(\Phi) \in \mathbb{R}^{N \times kD}$. Since each SRM projection costs $\mathcal{O}(N^2 \log N)$, the total cost across $k$ independent sketches is $\mathcal{O}(kN^2 \log N)$. Combining both stages, the overall runtime complexity of SRF is:

$$\mathcal{O}(NFD + kN^2 \log N).$$

This is asymptotically more efficient than using dense Gaussian projections, which incur a cost of $\mathcal{O}(kN^3)$, while preserving similar guarantees on distance preservation and kernel approximation [15].

Finally, in the context of our broader complexity analysis, which considers scaling with respect to the number of nodes $N$ and the number of training epochs $T$, we observe that SRF features are computed once at initialization and reused throughout training. Thus, like spectral encodings, SRF contributes a one-time cost and does not scale with $T$. The dominant term is the ROM projection, yielding an effective runtime complexity of $\mathcal{O}(N^2 \log N)$ and memory complexity of $\mathcal{O}(N)$ with respect to the graph size.

**Additional Discussion on Computational Complexity.** We analyze the computational and memory complexity of baseline methods described in Sections 3 and 4. Table 4 presents the complexity of these methods with respect to the number of nodes $N$. The analysis distinguishes between methods that require preprocessing versus those that perform computations during each forward pass. Preprocessing methods like random id [1] and SRF (ours) incur their computational cost once before training, then impose minimal overhead per epoch, while other methods perform their stated complexity computations during each forward pass. In scenarios where graphs are processed over many training epochs, preprocessing approaches achieve computational advantages as the number of epochs approaches or exceeds the graph size. Many graph learning benchmarks commonly found in the literature, including those used to evaluate PEARL and SPE, involve small graphs processed over many training epochs, favoring preprocessing approaches. Runtime benchmarks in Figure 2 complement this complexity analysis, demonstrating that SRF achieves superior computational efficiency in such scenarios. This efficiency advantage is also partially attributed to PEARL's requirement for a large number of graph samples that we empirically observe often scale with the number of graphs in the dataset. In scenarios where the required sample count is substantially smaller than the number of nodes, PEARL's empirical runtime would likely be comparatively more favorable. Future work should empirically validate this hypothesis.

Table 4: Comparison of computational and memory complexity for various GNN positional encoding methods with respect to number of nodes $N$. Methods marked with $*$ incur costs once before training, while others incur costs per forward pass. When the number of training epochs is comparable to graph size, preprocessing methods achieve computational advantages.

| Method | Computational Complexity | Memory Complexity |
|---|---|---|
| Random id$^*$ | $\mathcal{O}(N)$ | $\mathcal{O}(N)$ |
| SignNet | $\mathcal{O}(N^3)$ | $\mathcal{O}(N^2)$ |
| SignNet-8S | $\mathcal{O}(N^3)$ | $\mathcal{O}(N)$ |
| SignNet-8L | $\mathcal{O}(N)$ | $\mathcal{O}(N)$ |
| BasisNet | $\mathcal{O}(N^3)$ | $\mathcal{O}(N^2)$ |
| SPE | $\mathcal{O}(N^3)$ | $\mathcal{O}(N^2)$ |
| SPE-8S | $\mathcal{O}(N^3)$ | $\mathcal{O}(N^2)$ |
| R-PEARL | $\mathcal{O}(N)$ | $\mathcal{O}(N)$ |
| B-PEARL | $\mathcal{O}(N^2)$ | $\mathcal{O}(N)$ |
| SRF (ours)$^*$ | $\mathcal{O}(N^2 \log N)$ | $\mathcal{O}(N)$ |

## A.3 Additional Discussion on Baselines

For completeness, we describe the key methodological components of baseline encoding approaches referenced throughout the main manuscript. A qualitative summary of the differences between different baseline methods and SRF is provided in Table 5.

**SignNet and BasisNet.** Lim et al. [45] present SignNet and BasisNet, neural architectures that solve the sign and basis ambiguity problems in spectral graph positional encodings. SignNet addresses sign invariance by parameterizing functions of the form $f(v_1, \ldots, v_k) = \rho\left([\phi(v_i) + \phi(-v_i)]_{i=1}^k\right)$, where the structure $\phi(v_i) + \phi(-v_i)$ ensures invariance to sign flips of each eigenvector $v_i$, and $\phi$ and $\rho$ are neural networks (e.g., GIN and MLP respectively). BasisNet handles basis invariance in higher-dimensional eigenspaces by computing $f(V_1, \ldots, V_l) = \rho\left([\text{IGN}_{d_i}(V_i V_i^T)]_{i=1}^l\right)$, where $V_i \in \mathbb{R}^{n \times d_i}$ are orthonormal bases of eigenspaces, the mapping $V \mapsto VV^T$ produces the orthogonal projector which is invariant to basis changes $VQ$ for orthogonal $Q$, and IGNs (Invariant Graph Networks) process the resulting matrices while maintaining permutation equivariance. Both methods can incorporate eigenvalues and node features as additional inputs, and the processed eigenvectors are concatenated with original node features before being fed to downstream prediction models. Kanatsoulis et al. [36] introduce efficient variants: SignNet-8S and BasisNet-8S utilize the 8 smallest eigenvalues while maintaining $O(N^3)$ computational complexity, with SignNet-8S achieving reduced $O(N)$ memory complexity compared to the original $O(N^2)$. SignNet-8L uses the 8 largest eigenvalues, achieving both $O(N)$ computational and memory complexity.

**Hybrid Models and Miscellanies.** GPS [58] proposes a modular Graph Transformer framework that decouples local message-passing from global attention, combining three components: positional/structural encodings, local aggregation mechanisms, and global attention in a unified architecture. SAN [41] uses a learned positional encoding based on the Laplacian spectrum, which is added to node features before processing with a fully-connected Transformer. GNN-AK+ [71] extends standard MPNNs by replacing star-pattern aggregation with general subgraph pattern aggregation, where each node representation is computed from an induced subgraph encoding rather than just immediate neighbors. Graph ViT [29] adapts the Vision Transformer architecture, achieving linear complexity while capturing long-range dependencies and mitigating over-squashing through global receptive fields. SUN [24] combines subgraph-based methods with neural networks to enhance expressivity beyond traditional message-passing limitations.

## B GNN Architecture Analysis

To verify that our approach generalizes across different GNN architectures, we evaluate SRF with three additional backbone architectures: Graph Convolutional Networks (GCN) [40], Graph Attention Networks (GAT)[66], and Graph Attention Networks v2 (GATv2)[10]. We compare against random

Table 5: Qualitative comparison of SRF to common baselines. Properties summarize typical behavior discussed in the main text and Appendix. Entries consolidate theoretical and empirical properties established or referenced throughout the manuscript. $^{\dagger}$Denotes families of methods; specific variants may differ in details, but the stated properties reflect their typical guarantees and practical behavior.

| Method | Unique Representation? | Distance Sensitive? | Invariant or Equivariant? | Mitigates Oversquashing? | Alleviates Oversmoothing? |
|---|---|---|---|---|---|
| Random Node Features | Almost surely | No | In expectation | No | No |
| Spectral Encodings | No | Yes | Yes | No | No |
| Subgraph encodings$^{\dagger}$ | No | No | Yes | No | Unclear |
| Homomorphism counts$^{\dagger}$ | No | No | Yes | No | Unclear |
| PEARL | Unclear | With high probability | Yes | No | No |
| **SRF (Ours)** | **Almost surely** | **With high probability** | **In expectation** | **Yes** | **Yes** |

feature injection baselines that maintain the same total number of learnable parameters. Following Abboud et al. [1], we inject random features, but unlike their approach, we perform reinjection at each layer to provide a fair comparison with our method's parameter count. All experiments use the hyperparameter configurations of the GIN/E models for which results are presented in Section 4.

Table 6 presents the results on REDDIT-B and REDDIT-M datasets. Across all architectures, most SRF variants consistently outperform random feature baselines, demonstrating that the performance gains are not dependent on the specific choice of GNN backbone. However, we find that random features do slightly outperform a single variant, $(\mathcal{E}_{\text{RBF}}, \mathcal{S}_{\text{AG}}^{(8)})$ on one experiment (REDDIT-B dataset and GAT backbone). Interestingly, relative performance differences between SRF variants across architectures, are slightly different than in the trend found in Section 4.2, with $(\mathcal{E}_{\text{linear}}, \mathcal{S}_{\text{AG}}^{(8)})$ achieving slightly stronger comparative performance for some dataset/backbone pairs. Future work should investigate whether these performance variations arise from architecture-specific hyperparameter sensitivities or inherent compatibility differences between SRF variants and GNN backbones.

## C   Experimental Details

### C.1   Dataset Descriptions

Here, we describe properties of the real world graph datasetrs used in our experiments. All datasets are used in accordance with their respective licenses and terms of use.

**Reddit-B and Reddit-M.** The Reddit datasets [69] consist of graph classification tasks where each graph represents an online discussion thread. Nodes correspond to users and edges indicate response relationships between users' comments. Reddit-B contains 2,000 graphs across 2 classes with an average of 429.6 nodes per graph, where the task is to classify discussion threads as belonging to question/answer-based communities versus discussion-based communities. Reddit-M contains 5,000 graphs across 5 classes with an average of 508.5 nodes per graph, where the task is to predict which specific subreddit a discussion thread belongs to.

**DrugOOD.** The DrugOOD dataset [33] evaluates models on out-of-distribution generalization for drug-target binding affinity prediction. The dataset focuses on domain shifts arising from different bioassays (Assay), molecular scaffolds (Scaffold), and molecular sizes (Size), testing the ability to generalize to unseen experimental conditions, molecular structures, and compound sizes respectively. Each sample consists of paired protein-compound data with binary classification labels indicating binding activity.

**Peptides-struct.** The Peptides-struct dataset [22] from the Long Range Graph Benchmark comprises over 15,000 molecular graphs with more than 2 million nodes total, where individual graphs range from 8 to 444 nodes. The dataset involves multi-label regression on 3D structural properties of peptides, including inertia, length, sphericity, and geometric fit measures. Graphs are constructed with heavy atoms as nodes and chemical bonds as edges, requiring models to capture long-range interactions without explicit 3D coordinate information.

Table 6: Performance (Accuracy %) comparison of SRF variants across different GNN architectures on Reddit datasets. Random feature injection is a parameter-matched baseline with reinjection at each layer.

| Architecture | Method | REDDIT-M | REDDIT-B |
|---|---|---|---|
| GCN | Random Features | $53.46 \pm 1.00$ | $91.60 \pm 1.04$ |
| | $(\mathcal{E}_{\text{linear}}, \mathcal{S}_{\text{AG}}^{(8)})$ | $58.04 \pm 1.00$ | $93.30 \pm 0.72$ |
| | $(\mathcal{E}_{\mathcal{L}}, \mathcal{S}_{\text{AG}}^{(8)})$ | $57.20 \pm 0.90$ | $93.30 \pm 0.72$ |
| | $(\mathcal{E}_{\text{RBF}}, \mathcal{S}_{\text{AG}}^{(8)})$ | $58.08 \pm 0.89$ | $93.05 \pm 0.16$ |
| GAT | Random Features | $47.75 \pm 0.84$ | $86.55 \pm 1.89$ |
| | $(\mathcal{E}_{\text{linear}}, \mathcal{S}_{\text{AG}}^{(8)})$ | $51.40 \pm 1.04$ | $89.05 \pm 1.17$ |
| | $(\mathcal{E}_{\mathcal{L}}, \mathcal{S}_{\text{AG}}^{(8)})$ | $49.64 \pm 0.96$ | $88.15 \pm 1.43$ |
| | $(\mathcal{E}_{\text{RBF}}, \mathcal{S}_{\text{AG}}^{(8)})$ | $49.72 \pm 1.02$ | $86.15 \pm 1.43$ |
| GATv2 | Random Features | $46.88 \pm 0.89$ | $84.2 \pm 0.76$ |
| | $(\mathcal{E}_{\text{linear}}, \mathcal{S}_{\text{AG}}^{(8)})$ | $52.04 \pm 0.56$ | $87.4 \pm 0.74$ |
| | $(\mathcal{E}_{\mathcal{L}}, \mathcal{S}_{\text{AG}}^{(8)})$ | $48.76 \pm 0.89$ | $87.6 \pm 0.42$ |
| | $(\mathcal{E}_{\text{RBF}}, \mathcal{S}_{\text{AG}}^{(8)})$ | $48.40 \pm 1.27$ | $86.7 \pm 1.30$ |

## C.2 Training Details and Hyperparameter Search Procedure

All experiments use SRF with parameters $k = 8$ and search over the SRF hyperparameter $D \cdot k \in \{16, 32, 64, 128, 256\}$ (See Section 3). Hyperparameter optimization is conducted using Weights and Biases across all datasets. The Adam optimizer is used throughout all experiments.

**Reddit Datasets.** Following standard practice [36], we employ 10-fold cross-validation and report results for the best epoch across 330 training epochs. We report mean and standard deviation across folds. We use CrossEntropy loss with sum pooling. The hyperparameter grid search includes: number of layers $\in \{3, 4, 5, 6, 7, 8\}$, hidden dimensions $\in \{32, 64, 128, 256, 512\}$, batch size $\in \{16, 32, 64, 128\}$, and learning rate $\in [10^{-5}, 10^{-2}]$.

**DrugOOD.** We follow the experimental setup from [36], training for 150 epochs and reporting results on the out-of-distribution test set using L1 loss. The hyperparameter search includes: number of layers $\in \{3, 4, 5, 6, 7\}$, batch size $\in \{16, 32, 64, 128\}$, learning rate $\in [10^{-5}, 10^{-2}]$, layer normalization $\in \{\text{true}, \text{false}\}$, batch normalization $\in \{\text{true}, \text{false}\}$, and hidden dimensions $\in \{32, 64, 80, 90, 100, 110\}$.

**Peptides-struct.** Models are trained for 500 epochs using L1 loss with residual connections. The hyperparameter search includes: number of layers $\in \{4, 5, 6, 7, 8, 9\}$, hidden dimensions $\in \{70, 95, 105, 135, 150\}$, batch size $\in \{10, 25, 50, 75\}$, learning rate $\in [10^{-5}, 10^{-2}]$, layer normalization $\in \{\text{true}, \text{false}\}$, and batch normalization $\in \{\text{true}, \text{false}\}$. We additionally include the hyperparameters used by PEARL [36] in their evaluation, as we combine the methods (Section 4). Results are reported for $\mathcal{E}_{\text{linear}}$ sketch variant. Error bars as reported by Kanatsoulis et al. [36] are $\pm 0.001$ for PEARL-B, PEARL-R, GPS, SUN, and SAN+RWSE; $\pm 0.002$ for Graph ViT; $\pm 0.004$ for SAN+LapPE; and $0.00$ for GNN-AK+. The error bars for our variants are $\pm 0.004$.

## C.3 Hardware and Software Tools

Our graph processing and learning experiments utilize the open-source PyTorch Geometric library as the primary framework, with NetworkX serving as a supplementary tool for graph operations. To ensure reproducibility and accurate runtime evaluation, we include a catalog of all software dependencies and their specific versions in the Supplementary material. An anonymized implementation of our codebase is also made available in the Supplementary material. All performance evaluations were conducted using an AMD EPYC 7713 64-Core Processor running Red Hat Enterprise Linux 9.3

and a NVIDIA DGX A100 GPUs (80GB memory). At times during experimentation, a cluster of 8 such GPUs were used to run parallel experiments.

## C.4 Limitations

While SRF provides a principled and efficient mechanism for enhancing GNNs with global, feature-based information, it also has several limitations. First, its effectiveness relies on the presence of informative node features; in domains where features are sparse, noisy, or absent, performance gains may be limited. Second, as SRF relies on randomized projections, its properties hold in expectation; in practice, this introduces variance across runs, although our empirical results indicate this effect is negligible. Third, SRF is inherently topology-agnostic and may fail to capture structural signals that positional encodings or spectral methods explicitly model, but as noted, SRF can be used to complement these approaches. Finally, SRF introduces modest computational overhead compared to a vanilla GNN, though it is much more efficient than spectral methods.

# D  Analysis of SRF Embedding Dimension and Projection Count

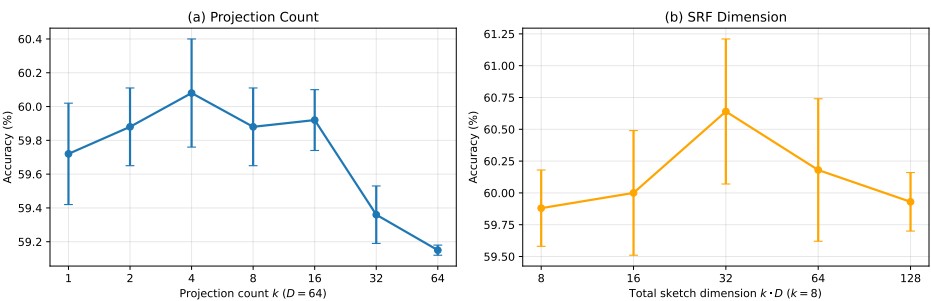

Figure 4: Hyperparameter sensitivity analysis on REDDIT-M using $(\mathcal{E}_{\text{linear}}, S_{AG}^{(k)})$. (a) Performance (accuracy %) vs. projection count $k$ with $D = 64$ fixed. (b) Performance (accuracy %) vs. total sketch dimension $k \cdot D$ with $k = 8$ fixed.

The SRF framework introduces two hyperparameters that control the quality and computational cost of the sketched embeddings: embedding dimension $D$ and projection count $k$ 3. Understanding the sensitivity of SRF performance to these parameters is helpful for practical deployment.

The embedding dimension $D$ governs the fidelity of the underlying kernel approximation as established in the main manuscript. Larger $D$ provides more accurate kernel estimates but increases computational overhead. The projection count $k$ determines the rank approximation of the random projection, with higher $k$ potentially improving the quality of cross-node information encoding at the cost of increased overhead.

In this Appendix, we evaluate the impact of these hyperparameters with a simple case-study experiment. To isolate the effect of projection count $k$, we fix $D = 64$ and vary $k \in \{1, 2, 4, 8, 16, 32, 64\}$. To assess the impact of embedding dimension $D$ while controlling for total sketch dimensionality, we fix the total sketched feature size $k \cdot D \in \{8, 16, 32, 64, 128\}$ and set $k = 8$, allowing $D$ to vary accordingly.

Our hyperparameter sensitivity results are presented in Figure 4 for the REDDIT-M dataset using operator $(\mathcal{E}_{\text{linear}}, S_{AG}^{(k)})$. Due to computational limitations, we focus the analysis on this single dataset-operator combination.

**Projection Count Analysis (Figure 4a).** The results demonstrate a performance trade-off when varying the projection count $k$ and maintaining fixed total sketch dimensionality. Performance peaks at $k = 4$ and subsequently declines as $k$ increases. This pattern is likely due to an underlying trade-off between the number of independent projections and the quality of said individual projections: with fixed $D = 64$, increasing $k$ provides more diverse cross-node information but at the cost of lower-dimensional kernel feature representations. The decline in performance at high $k$ values suggests

that the degradation in kernel approximation quality eventually outweighs the benefits of additional projections.

**SRF Dimension Analysis (Figure 4b).** When varying the total sketch dimension $k \cdot D$ with fixed $k = 8$, we observe a modest performance increase from $k \cdot D = 8$ to $k \cdot D = 64$, followed by diminishing returns at higher dimensions. However, the substantial variance across runs suggest these differences may not be statistically significant. The performance decline at $k \cdot D = 128$ indicates that excessive sketch dimensionality may lead to overfitting on this moderately-sized dataset. Future work should investigate scaling properties more thoroughly on larger datasets where overfitting concerns are mitigated.

**Practical Guidance on Hyperparameter Tuning.** Based on our empirical study of SRF hyperparameters and the complexity analysis, we provide the following practical guidance for selecting SRF configurations. For kernel selection, we recommend RBF embeddings $\mathcal{E}_{\mathrm{RBF}}$ when computational constraints prevent extensive kernel sweeps. The projection count $k$ typically exhibits an inverted-U effect on performance: accuracy improves up to a point before declining; across evaluated datasets we find $k = 4$ to be a reasonable default. Finally, increasing the total sketch dimension $k \cdot D$ improves performance with linear memory/runtime cost but shows diminishing returns; we suggest $k \cdot D = 64$ as a practical starting point, with adjustments based on dataset size and available compute.

