# OpenReview forum: "Sketch-Augmented Features Improve Learning Long-Range Dependencies in Graph Neural Networks"
_NeurIPS.cc/2025/Conference — NeurIPS 2025 poster_

### Official Review · Reviewer_5P6B · 2025-07-02

**Clarity:** 3
**Significance:** 2
**Originality:** 2
**Rating:** 4
**Confidence:** 3

**Summary:**

GNNs often face challenges such as oversquashing, oversmoothing, and limited expressiveness, particularly when modeling long-range dependencies. This paper introduces Sketched Random Features (SRF), randomized global summaries of node features designed to address these issues. SRF is created by first projecting node features into a kernel space using random mappings such as random Fourier features, and then aggregating them across nodes via the Johnson–Lindenstrauss transform. The resulting sketches are unique, distance-sensitive, and independent of graph topology. By injecting these representations into each GNN layer, the model gains efficient access to global context without relying on costly attention mechanisms. This approach consistently improves performance on both synthetic and real-world graph benchmarks, while offering significant gains in memory and runtime efficiency over transformer-based methods.

**Questions:**

I would request the authors to address the points raised in the weakness section.

**Ethical Concerns:**

["NO or VERY MINOR ethics concerns only"]

**Final Justification:**

The authors have addressed my concerns, and I have raised my score accordingly. However, I still believe that the theoretical analysis, while well-grounded, could have been more novel. For this reason, I have leaned toward a borderline accept.

**Limitations:**

The societal or ethical implications of the work are not deeply discussed. For instance, while the method is efficient and scalable, are there settings such privacy-sensitive or fairness-constrained healthcare applications where injecting global summaries might amplify biases?

**Paper Formatting Concerns:**

No formatting issues

**Quality:**

3

**Strengths And Weaknesses:**

The paper is clearly written and easy to follow. It presents the core limitations of GNNs in a well-motivated and accessible manner. The proposed method is grounded in well-established theoretical foundations and makes effective use of them. Empirical results demonstrate that incorporating SRF improves both performance and efficiency compared to strong baselines.

Despite these positive points, I have the some reservations:

 - SRF is positioned as a topology-agnostic method. It is entirely based on node features and random projections, which may reduce effectiveness in tasks where structural cues are critical. While this is acknowledged as complementary to structural encodings, the standalone method may underperform in structurally complex graphs with weak features.

 - SRF relies on random kernel mappings and projections. This may lead to variability in performance across different runs, especially in low-data or low-variance settings. The paper does not deeply investigate the robustness of SRF across seeds or noise levels.

 - There is no detailed ablation on sketching parameters such as embedding dim and kernel choice.

 - While the method is well motivated and grounded in established theoretical tools such as random kernel features and the JL transform, it does not introduce any novel theoretical results. The analysis largely restates known properties and lacks deeper contributions, such as convergence guarantees, generalisation bounds, or new theoretical insights into the behaviour of GNNs.

---

> ### Author Rebuttal · Authors · 2025-07-31
>
> Thanks for your thoughtful review and for highlighting the clear presentation, strong theoretical grounding, and empirical results of our work. We address your concerns individually below.
>
> > W1: SRF is positioned as a topology-agnostic method. It is entirely based on node features and random projections, which may reduce effectiveness in tasks where structural cues are critical. While this is acknowledged as complementary to structural encodings, the standalone method may underperform in structurally complex graphs with weak features.
>
> Our method is designed for the common setting where graphs contain informative node features, enabling alternatives to computationally expensive structural encodings. As acknowledged in our limitations (Appendix E.4, lines 750-752), performance gains may be limited when node features are sparse or noisy. However, as you note, SRF can complement structural encodings when needed, and in the worst case (featureless graphs), it reduces to random node individualization—an effective method in itself [1]. This can be seen by applying Algorithm 1 to graphs where all nodes have identical unit features.
>
> > W2: SRF relies on random kernel mappings and projections. This may lead to variability in performance across different runs, especially in low-data or low-variance settings. The paper does not deeply investigate the robustness of SRF across seeds or noise levels.
>
> We appreciate this important concern. Our experiments vary random seeds and report error bars across multiple runs, demonstrating statistically significant improvements that are often more stable than PEARL (a neural network-based positional encoding baseline). Regarding low-data regime, our empirical evaluation spans diverse scales, from small datasets like REDDIT-B (2,000 graphs) to larger-scale benchmarks, with consistent improvements across this range, suggesting robustness to dataset size variations.
>
> Could you please clarify what you mean by "low-variance settings"? We want to ensure we address your specific concern accurately.
>
> > W3: There is no detailed ablation on sketching parameters such as embedding dim and kernel choice.
>
> We do empirically analyze embedding dimension and kernel choice: We evaluate 4 kernel types across all experiments (Figures 1a-b, Tables 1-2) and examine embedding dimension/projection count effects in Appendix F. Please see our response to Reviewer hHzn W2.3/Q2.3 for a detailed summary of these results.
>
> > W4: While the method is well motivated and grounded in established theoretical tools such as random kernel features and the JL transform, it does not introduce any novel theoretical results. The analysis largely restates known properties and lacks deeper contributions, such as convergence guarantees, generalisation bounds, or new theoretical insights into the behaviour of GNNs.
>
> We respectfully disagree that our work lacks novel theoretical contributions. We provide five new propositions (Section 3.3) establishing fundamental properties specific to our sketching approach for GNNs. Additionally, we provide the first theoretical analysis connecting how sketched embeddings specifically mitigate oversquashing (through topology-agnostic information flow) and oversmoothing (through distance-sensitive uniqueness preservation). Our theoretical contributions focus on justifying SRF's effectiveness rather than general GNN behavior, which we consider appropriate for the scope of the manuscript. Please see our response to Reviewer hHzn W1/Q1 for additional discussion on the novelty of our approach more generally.
>
> > L1: The societal or ethical implications of the work are not deeply discussed. For instance, while the method is efficient and scalable, are there settings such privacy-sensitive or fairness-constrained healthcare applications where injecting global summaries might amplify biases?
>
> We do not believe our work introduces ethical implications beyond those inherent to improving ML algorithms generally. However, you raise an interesting point about privacy-sensitive applications. While privacy analysis is beyond our current scope, the sketching process does aggregate node information globally, which could potentially leak private information in sensitive domains. At the same time, sketching techniques are also foundational to many differential privacy mechanisms [2]. We believe such analysis is an interesting direction for future work.
>
> [1] Abboud, Ralph, et al. "The surprising power of graph neural networks with random node initialization." arXiv preprint arXiv:2010.01179 (2020).
>
> [2] Blocki, Jeremiah, et al. "The johnson-lindenstrauss transform itself preserves differential privacy." 2012 IEEE 53rd annual symposium on foundations of computer science. IEEE, 2012.

---

> > ### Comment · Reviewer_5P6B · 2025-08-02
> >
> > I thank the authors for their detailed response, which alleviates most of my concerns. I agree that the method is well-grounded in existing theoretical foundations, and the tailored propositions provide meaningful insights into the behaviour of SRF . While the paper may not introduce fundamentally novel theoretical contributions, this does not diminish the value of the work.
> >
> > I have revised my score accordingly.

---

### Official Review · Reviewer_i8dd · 2025-07-02

**Clarity:** 3
**Significance:** 3
**Originality:** 4
**Rating:** 5
**Confidence:** 3

**Summary:**

The authors propose a node augmentation method that individualizes node features while further enriching them with global information. This is accomplished by concatenating the original node features with randomized projections of kernel embeddings of the node features. These so-called 'Sketched Random Features' are computed by first approximating kernel embeddings on the node features, which are then transformed via a random projection matrix. Leveraging results from the Johnson–Lindenstrauss lemma, the authors derive theoretical guarantees that their method mitigates common GNN issues such as oversquashing and oversmoothing. In an experimental evaluation, the authors support their claims on oversquashing and oversmoothing and demonstrate that the individualized node features enable more expressive GNN models. Furthermore, the approach is validated on standard graph classification benchmarks.

**Questions:**

- How are graphs of varying sizes handled? If I understand correctly, both the embedding and sketch operators assume a fixed graph size N.
- Individualisation of nodes generally comes at the cost of worse generalisation of the model. Do you have an intuition on how your method affects sample complexity?

**Ethical Concerns:**

["NO or VERY MINOR ethics concerns only"]

**Final Justification:**

The reviewers have addressed all my concerns and questions. I maintain my positive score.

**Limitations:**

yes

**Paper Formatting Concerns:**

No concerns.

**Quality:**

4

**Strengths And Weaknesses:**

Strengths:
- The article is well written and structured.
- The technical quality of the paper is excellent.
- Although conceptually simple, the approach is elegant in how it simultaneously addresses two key aspects: global node information propagation and node individualization.
- The theoretical results are presented clearly and appropriately support the paper’s claims.
- The benchmarks on oversmoothing and oversquashing are convincing.

Weaknesses:
- While the title mentions long-range dependencies, this aspect is not much explored in depth within the paper. Maybe the authors can evaluate their method on standard long-range benchmark datasets.
- Minor notes:
	- Algorithm 1 is placed in Section 1 but not discussed until Section 3 (page 6). Thus, some of the notations have not been introduced yet such as \Epsilon or \mathcal{S}_AG.

---

> ### Author Rebuttal · Authors · 2025-07-31
>
> Thank you for your positive review and for recognizing the overall technical quality, approach to addressing global information propagation/node individualization, theoretical results, and comprehensive empirical evaluation of our work. We address your concerns and questions individually below.
>
> > W1: While the title mentions long-range dependencies, this aspect is not much explored in depth within the paper. Maybe the authors can evaluate their method on standard long-range benchmark datasets.
>
>  We evaluate the ability of Sketched Random Features to account for long-range dependencies in both synthetic and real-world scenarios. The synthetic Tree-NeighborsMatch dataset is specifically designed to measure oversquashing and the compression of long-range dependencies [1], where our results demonstrate strong performance (Figure 1a-b). Additionally, we evaluate on the peptides-struct dataset from the Long Range Graph Benchmark [2], which is designed to require reasoning about long-range dependencies in order to excel (lines 347-349, Table 3). Our performance on this real-world long-range benchmark further validates SRF's effectiveness on long-range dependency tasks. In response to your feedback, we have clarified the motivation for these dataset choices and their relevance to long-range dependency evaluation in the manuscript.
>
> > W2: Algorithm 1 is placed in Section 1 but not discussed until Section 3 (page 6). Thus, some of the notations have not been introduced yet such as \Epsilon or \mathcal{S}_AG.
>
> Thanks for pointing this out. We have added additional detail to the text so as to properly introduce notation that appears in Algorithm 1.
>
> > Q1: How are graphs of varying sizes handled? If I understand correctly, both the embedding and sketch operators assume a fixed graph size N.
>
> Our method naturally handles graphs of varying sizes (indeed, all our evaluated datasets contain graphs of different sizes). To see why, we note that the embedding operator $\mathcal{E}$ and sketch operator $\mathcal{S}$ take as input a matrix with $N$ rows, where $N$ is the number of nodes in the specific graph being processed. We use structured random matrices (SRM) as described in lines 271-276, which can be instantiated for any graph size at runtime by instantiating an appropriately sized SRM and indexing only the number of rows required for a given graph. This is similar to standard practice in the SRM literature [3] or as used to sketch dynamic graphs for representation [4].
>
> > Q2: Individualisation of nodes generally comes at the cost of worse generalisation of the model. Do you have an intuition on how your method affects sample complexity?
>
> We hypothesize that SRF has better sample complexity than naive random individualization. Unlike random features that create arbitrary coordinate spaces, SRF preserves meaningful geometric structure: our embeddings retain distance sensitivity in kernel space with high probability (Proposition 3.2), providing principled similarity measures between nodes. This distance preservation should facilitate generalization by maintaining inductive biases from the feature space. Empirically, our method performs well even on our smallest dataset (REDDIT-B with 2,000 graphs), suggesting reasonable sample efficiency. Future work should investigate formal sample complexity analysis.
>
> [1] Alon, Uri, and Eran Yahav. "On the Bottleneck of Graph Neural Networks and its Practical Implications." International Conference on Learning Representations.
>
> [2] Dwivedi, Vijay Prakash, et al. "Long range graph benchmark." Advances in Neural Information Processing Systems 35 (2022): 22326-22340.
>
> [3] Choromanski, Krzysztof M., Mark Rowland, and Adrian Weller. "The unreasonable effectiveness of structured random orthogonal embeddings." Advances in neural information processing systems 30 (2017).
>
> [4] Hosseini, Ryien, et al. "Quality Measures for Dynamic Graph Generative Models." The Thirteenth International Conference on Learning Representations.

---

> > ### Comment · Reviewer_i8dd · 2025-08-05
> >
> > Thank you very much for addressing all my points. I will maintain my positive score and continue to recommend acceptance of the article.

---

### Official Review · Reviewer_hHzn · 2025-07-02

**Clarity:** 3
**Significance:** 3
**Originality:** 3
**Rating:** 4
**Confidence:** 4

**Summary:**

The paper proposes Sketched Random Features (SRF), a method for augmenting message-passing Graph Neural Networks (GNNs) with global, randomized embeddings derived from node features. Combining kernel random features and Johnson–Lindenstrauss sketching, these embeddings are injected into each GNN layer to mitigate oversquashing, oversmoothing, and limited expressiveness. The paper provides theoretical justifications for the approach, and experiments on both synthetic and real-world benchmarks demonstrate improved performance and efficiency over standard GNNs and other positional encoding strategies.

**Questions:**

1. How does SRF  advance beyond prior random feature–based or positional encoding methods for GNNs?

2.1 Have the authors evaluated SRF with diverse GNN architectures (e.g., GraphSAGE, GAT, GCNII) to support its claimed architecture-agnostic applicability?

2.2 Have the authors evaluated on more popular datasets (e.g., from Open Graph Benchmark) and other baselines addressing oversmoothing and oversquashing?

2.3 How does SRF’s performance vary with respect to key hyperparameters such as sketch dimension, kernel type, or the number of random projections?

2.4. Can you provide practical guidance on selecting hyperparameters (e.g., sketch dimension) to balance model accuracy, distance preservation, and computational cost?

**Ethical Concerns:**

["NO or VERY MINOR ethics concerns only"]

**Final Justification:**

I have increased my score based on the author's response. I still believe that the novelty is limited; however, the results are good. I believe the writing has to improve significantly to clarify how the proposed work differs from existing work and improves upon it.

**Limitations:**

yes

**Quality:**

3

**Strengths And Weaknesses:**

## Strengths

- Clearly motivated solution to major GNN limitations
- Solid theoretical analysis of key properties
- Empirically effective on diverse benchmarks
- Efficient in runtime and memory
- Modular and conceptually elegant design

## Weaknesses

- *Novelty*: The core contribution combines well-understood ideas -- random feature maps and sketching -- rather than introducing fundamentally new algorithms or theoretical insights. While the combination works, the paper should clarify how SRF advances beyond prior random feature–based or positional encoding methods for GNNs.

- *Experiments*
  - *Limited architectural diversity in evaluation*: The experiments focus predominantly on GIN/GINE architectures. Although the method is described as architecture-agnostic, the paper does not assess integration or performance with other state-of-the-art models (e.g., GraphSAGE, GAT, GCNII). A broader empirical evaluation could validate SRF’s general applicability.

  - *Dataset/baseline choices*: There have been many papers that address the issues of oversquashing, oversmoothing, and expressivity. It would be better if the authors choose more datasets that are popular in the literature so that they can compare against a more diverse set of methods (e.g., see datasets and methods in the Open Graph Benchmark).

  - *Ablation and sensitivity analysis*: The manuscript briefly mentions variations such as kernel types, projection dimensions, and embedding strategies, but lacks thorough ablations examining how these choices impact empirical performance, stability, and runtime. For example, how sensitive are results to sketch dimension, kernel selection, or the number of random projections? Such analyses would offer practical guidance for applying SRF.

  - *Limited discussion of practical trade-offs*: The paper does not systematically discuss how to choose practical hyperparameters (especially sketch dimension) to balance distance preservation, efficiency, and model accuracy. Insights on this front would strengthen the work’s utility and reproducibility.

- *Presentation* (minor issue): The paper does not provide schematics or diagrams showing how SRF embeddings are computed or incorporated into GNN layers. A clear schematic would help readers understand the interplay between SRF embedding and message passing, especially for practitioners aiming to implement the method.

---

> ### Author Rebuttal · Authors · 2025-07-31
>
> Thanks for your review and for identifying the strong theoretical analysis, empirical effectiveness, and computational efficiency of our approach. We address your specific concerns below.
>
> > W1: Novelty: The core contribution combines well-understood ideas -- random feature maps and sketching -- rather than introducing fundamentally new algorithms or theoretical insights.  While the combination works, the paper should clarify how SRF advances beyond prior random feature–based or positional encoding methods for GNNs.
>
> > Q1: How does SRF advance beyond prior random feature–based or positional encoding methods for GNNs?
>
> SRF's novelty lies in being the first to use sketching for mixing global feature information in graphs rather than dimensionality reduction. Specifically, the use of this novel sketching application with random kernel features uniquely allows our approach to mix all node information at each layer, regardless of geodesic distance. Unlike prior methods, this combination yields signals that are provably (1) almost surely unique, (2) distance-sensitive in the feature space with high probability, and (3) permutation-equivariant in expectation (Propositions 3.1–3.5). These properties are not simultaneously provided by prior random-feature or positional encoding methods (summarized in the table below). As shown in our manuscript, these properties mitigate or overcome well-known shortcomings of message-passing GNNs (expressiveness, oversquashing, oversmoothing). Furthermore, to the best of our knowledge, our work is the first to leverage node features (rather than random or structure-based methods) to create encodings that mitigate these aforementioned shortcomings of GNNs.
>
> **We discuss how SRF advances beyond prior encoding methods throughout the manuscript.** For example, the introduction (lines 47-57) overviews how our feature-based approach overcomes the limitations of prior methods. The specific weaknesses of existing random/positional encoding methods are discussed in lines 141-151. Section 3.3 (lines 251-270) discusses in detail how specific analytic properties of our proposed approach are able to overcome the aforementioned issues. In response to your feedback, we further provide the following table summarizing the differences between the methods.
>
> | Method | Unique Representation? | Distance Sensitive? | Equivariant? | Mitigates Oversquashing? | Alleviates oversmoothing? |
> |--------|------------------------|-------------------|--------------|-------------------------|--------------------------|
> | Random Node Features | Almost surely | No | In expectation | No | No |
> | Spectral Encodings | No | Yes | Yes | No | No |
> | PEARL | Unclear | With high probability | Yes | No | No |
> | SRF (Ours) | Almost surely | With high probability | In expectation | Yes | Yes |
>
> > W2.1: Limited architectural diversity in evaluation: The experiments focus predominantly on GIN/GINE architectures. Although the method is described as architecture-agnostic, the paper does not assess integration or performance with other state-of-the-art models (e.g., GraphSAGE, GAT, GCNII). A broader empirical evaluation could validate SRF’s general applicability.
> > Q2.1: Have the authors evaluated SRF with diverse GNN architectures (e.g., GraphSAGE, GAT, GCNII) to support its claimed architecture-agnostic applicability?
>
> As described in our experimental overview (lines 278-288), while we present results for GIN/GINE in the main manuscript, we evaluate SRF with other GNN architectures in the Appendix (found in our supplementary material). **Appendix D presents results on the GCN, GAT, and GATv2 architectures across two datasets (Table 5), demonstrating consistent improvements over baselines and confirming SRF's architecture-agnostic benefits.**
>
> > W2.2: Dataset/baseline choices: There have been many papers that address the issues of oversquashing, oversmoothing, and expressivity. It would be better if the authors choose more datasets that are popular in the literature so that they can compare against a more diverse set of methods (e.g., see datasets and methods in the Open Graph Benchmark).
> > Q 2.2: Have the authors evaluated on more popular datasets (e.g., from Open Graph Benchmark) and other baselines addressing oversmoothing and oversquashing?
>
> Our evaluation comprises of 6 real-world and 3 synthetic datasets, targeting the core GNN limitations SRF addresses. We evaluate comprehensive baselines spanning multiple classes of positional encodings: random features, spectral embeddings, canonized variants, and PEARL (recent state-of-the-art). Our datasets cover diverse domains (e.g. social networks, molecules), tasks (e.g. classification, regression), out-of-distribution generalization (DrugOOD dataset), and GNN backbone architectures (GIN, GINE, GCN, GAT, GATv2). Importantly, our evaluation includes datasets specifically designed for the phenomena our method addresses: oversquashing is evaluated both synthetically (Tree-NeighborsMatch) and on a real-world dataset (peptides-struct from the Long Range Graph Benchmark), while oversmoothing and expressiveness are assessed through targeted synthetic benchmarks. *Additionally, the datasets used are standard in the literature, allowing easier comparison with several baselines (e.g. see recent baselines e.g. [1, 2, 3, 4] that evaluate on overlapping subsets of these same datasets).* We welcome specific suggestions for additional evaluation areas if you have particular concerns.
>
> > W2.3: Ablation and sensitivity analysis: The manuscript briefly mentions variations such as kernel types, projection dimensions, and embedding strategies, but lacks thorough ablations examining how these choices impact empirical performance, stability, and runtime. For example, how sensitive are results to sketch dimension, kernel selection, or the number of random projections? Such analyses would offer practical guidance for applying SRF.
> > Q 2.3: How does SRF’s performance vary with respect to key hyperparameters such as sketch dimension, kernel type, or the number of random projections?
>
> **We provide comprehensive empirical analyses of the noted hyperparameters throughout the manuscript and appendices.** We evaluate on 4 kernel types (3 normal + ablation kernel) in Section 4 (e.g. Figures 1a-1b, Tables 1-2). As noted in our experimental overview (lines 286-288), we evaluate the effect of varying embedding dimension $k\cdot D$ and projection count $k$ in Appendix F. We summarize some of these results/key takeaways again for reference:
> - For many real-world graphs, the RBF embedding $\mathcal{E}_{RBF}$ has the best empirical performance among kernel types, though this result is not always statistically significant. (See Table 2)
> - Projection count: For Reddit datasets, performance peaks at $k=4$, then declines due to the trade-off between projection diversity and individual projection quality (Appendix, Figure 3a).
> - Sketch dimension: Modest improvements from $k \cdot D=8$ to $k \cdot D=64$, with diminishing returns thereafter (Appendix, Figure 3b).
>
> > W2.4: Limited discussion of practical trade-offs: The paper does not systematically discuss how to choose practical hyperparameters (especially sketch dimension) to balance distance preservation, efficiency, and model accuracy. Insights on this front would strengthen the work’s utility and reproducibility.
> > Q 2.4: Can you provide practical guidance on selecting hyperparameters (e.g., sketch dimension) to balance model accuracy, distance preservation, and computational cost?
>
> Based on our empirical study of SRF hyperparameters (Appendix F) and complexity analysis (lines 271-276, Appendix A), we provide the following practical guidance (to be added to Appendix E):
> - Kernel selection: Use RBF embedding $\mathcal{E}_{RBF}$ when computational constraints prevent extensive kernel comparison.
> - Projection count: Performance typically follows an inverted-U pattern, peaking then declining with increasing $k$. We find $k=4$ provides a robust default across our evaluated datasets.
> - Sketch dimension: Increasing $k \cdot D$ improves performance with linear memory/runtime costs, but exhibits diminishing returns. We recommend $k \cdot D=64$ as a practical starting point, with adjustment based on dataset size and computational budget.
>
> > W3: Presentation (minor issue): The paper does not provide schematics or diagrams showing how SRF embeddings are computed or incorporated into GNN layers. A clear schematic would help readers understand the interplay between SRF embedding and message passing, especially for practitioners aiming to implement the method.
>
> We agree that a schematic would improve clarity and have prepared a diagram illustrating SRF computation and injection into GNN layers (visualizing Algorithm 1). Due to rebuttal constraints preventing figure uploads, we will include this in a possible camera-ready version.
>
> [1] Kanatsoulis, Charilaos, et al. "Learning Efficient Positional Encodings with Graph Neural Networks." The Thirteenth International Conference on Learning Representations.
>
> [2] He, Xiaoxin, et al. "A generalization of vit/mlp-mixer to graphs." International conference on machine learning. PMLR, 2023.
>
> [3] Huang, Yinan, et al. "On the Stability of Expressive Positional Encodings for Graphs." The Twelfth International Conference on Learning Representations.
>
> [4] Finkelshtein, Ben, et al. "Cooperative Graph Neural Networks." International Conference on Machine Learning. PMLR, 2024.

---

> > ### Comment · Reviewer_hHzn · 2025-08-06
> >
> > Thanks for the detailed response. I have improved my score. However, I would recommend making improvements in writing -- particularly contrasting the proposed work with the existing work. Currently, it seems a bit disconnected, making it hard to assess novelty
> >
> > > For example, the introduction (lines 47-57) overviews how our feature-based approach overcomes the limitations of prior methods. The specific weaknesses of existing random/positional encoding methods are discussed in lines 141-151. Section 3.3 (lines 251-270)
> >
> > It would be helpful if you could show or expand the table in your response to contrast against the methods in 141-151.
> >
> > Regarding the baselines, I understand that you have chosen these based on recent methods. But since oversmoothing and expressivity have been studied for a while, I am curious to see some older methods, for example, subgraph-based methods (shadowGNN [1] ). Perhaps something to consider in the future.
> >
> > [1] Zeng et. al, Decoupling the Depth and Scope of Graph Neural Networks, NeurIPS 2021.

---

> ### Author Response · Authors · 2025-08-09
>
> Thank you for your thoughtful response, for taking the time to reassess our work, and for the suggestions to improve clarity.
> Following your suggestion, we will move an expanded comparison table covering all categories listed in lines 141–151 (with representative citations) into the revised manuscript to make the novelty contrast more explicit. Here is a draft of the updated table for reference:
>
> | **Method** | **Unique Representation?** | **Distance Sensitive?** | **Invariant or Equivariant?** | **Mitigates Oversquashing?** | **Alleviates oversmoothing?** |
> |------------|----------------------------|-------------------------|----------------------------|-------------------------------|--------------------------------|
> | Random Node Features [1] | Almost surely | No | In expectation | No | No |
> | Spectral Encodings [2] | No | Yes | Yes | No | No |
> | Subgraph encodings [3]† | No | No | Yes | No | Unclear |
> | Homomorphism counts [4]† | No | No | Yes | No | Unclear |
> | PEARL [5] | Unclear | With high probability | Yes | No | No |
> | SRF (Ours) | Almost surely | With high probability | In expectation | Yes | Yes |
>
> †These categories encompass diverse families of methods and thus the entries reflect properties of the representative citations. Comprehensive analysis of variants within each family will be provided in Appendix A.1.
>
> Also based on your feedback, we will expand the discussion of prior encoding approaches in Appendix A.1 and include additional analysis of the works you suggested (including ShadowGNN) in the camera-ready.
>
> [1] Dasoulas, George, et al. "Coloring graph neural networks for node disambiguation." arXiv preprint arXiv:1912.06058 (2019).
>
> [2] Dwivedi, Vijay Prakash, et al. "Benchmarking graph neural networks." Journal of Machine Learning Research 24.43 (2023): 1-48.
>
> [3] Bouritsas, Giorgos, et al. "Improving graph neural network expressivity via subgraph isomorphism counting." IEEE Transactions on Pattern Analysis and Machine Intelligence45.1 (2022): 657-668.
>
> [4] Bao, Linus, et al. "Homomorphism Counts as Structural Encodings for Graph Learning." arXiv preprint arXiv:2410.18676 (2024).
>
> [5] Kanatsoulis, Charilaos I., et al. "Learning efficient positional encodings with graph neural networks." arXiv preprint arXiv:2502.01122 (2025).

---

### Official Review · Reviewer_xCTo · 2025-07-03

**Clarity:** 3
**Significance:** 3
**Originality:** 3
**Rating:** 5
**Confidence:** 3

**Summary:**

This paper studies graph learning with GNNs, and proposed a simple feature augmentation method to improve the long-range dependency learning of GNNs. The main idea is to create a new "sketch" for each node by taking all the node features in the graph and mixing them together using a designed but randomized projection. This sketch gives every node a unique, global signature that helps the GNN see long-range patterns and keep node embeddings from becoming too similar. The authors show that this simple and efficient trick works well on its own and can even be stacked with other methods to make GNNs more powerful.

**Questions:**

1. I would appreciate some discussion from the authors on how the proposed method would work on inductive systems where new nodes/edges show up frequently.

**Ethical Concerns:**

["NO or VERY MINOR ethics concerns only"]

**Final Justification:**

i appreciate the authors' rebuttal, and will keep my positive score

**Quality:**

3

**Strengths And Weaknesses:**

s1. This paper is overall clearly written.

s2. The proposed method, although complicate in design and analysis, is very easy to implement. The proposed method is also task agnostic.

s3. This paper is backed with good thoretical grounding. The authors provide formal propositions to explain why SRF works, showcasing that these sketches are unique, distance-sensitive, and contain cross-node information.

w1. The evaluation is a bit lightweight. This work can benefit from some more comprehensive experiments.

w2. The efficiency comparison in Fig 2 can also be extended to include more methods, which would make the comparison more convincing.

---

> ### Author Rebuttal · Authors · 2025-07-31
>
> Thanks for your thoughtful review of our work and for highlighting the clear presentation, theoretical grounding, and practicality of our work. We address your concerns and questions individually below.
>
> > W1. The evaluation is a bit lightweight. This work can benefit from some more comprehensive experiments.
>
> Our evaluation spans 6 real-world and 3 synthetic datasets across diverse domains (e.g. social networks, molecules), tasks (e.g. classification, regression), out-of-distribution generalization (DrugOOD dataset), and GNN backbone architectures (GIN, GINE, GCN, GAT, GATv2). Additionally, our evaluation is designed to target the core GNN limitations SRF addresses: oversquashing, oversmoothing, and expressiveness. Oversquashing is evaluated both synthetically (Tree-NeighborsMatch) and on a real-world dataset (Peptides-struct from the Long Range Graph Benchmark, which specifically tests long-range dependency modeling [1]). Oversmoothing and expressiveness are assessed through targeted synthetic benchmarks designed to isolate these phenomena. We welcome specific suggestions for additional evaluation areas if you have particular concerns.
>
> > W2. The efficiency comparison in Fig 2 can also be extended to include more methods, which would make the comparison more convincing.
>
> Figure 2 complements our theoretical complexity analysis (Appendix A.2, Table 4), which shows most baseline methods have unfavorable scaling with respect to graph size. We focus on PEARL since it has the most favorable theoretical complexity among baselines. In response to your feedback, we have clarified this in the manuscript (line 317): "on evaluated datasets, they have about 3 times faster runtime and use about two orders of magnitude less memory than PEARL, which is the baseline encoding with better asymptotic complexity (Appendix A.2)."
>
> > Q1. I would appreciate some discussion from the authors on how the proposed method would work on inductive systems where new nodes/edges show up frequently.
>
> Our sketch operator $S_{AG}$ uses an $N \times N$ projection matrix where $N$ is the number of nodes. For systems where new nodes appear, one approach is to set $N_{max}$ as an upper bound on graph size. When processing graphs with $N < N_{max}$, the remaining $N_{max} - N$ positions correspond to zero-padded placeholder nodes. As new nodes appear, they can be assigned to these placeholder positions without changing the sketch matrix.
>
> [1] Dwivedi, Vijay Prakash, et al. "Long range graph benchmark." Advances in Neural Information Processing Systems 35 (2022): 22326-22340.

---

### Note · Authors · 2025-08-13

We’d like to thank the reviewers for taking the time to evaluate our work. We are happy that the reviewers received our work positively and appreciate the suggestions for improving the manuscript. We have fully addressed the reviewer concerns and questions during the rebuttal and discussion periods.

Below, we provide a brief summary of the changes made to the manuscript in response to feedback during these periods:
- We have clarified that a key novelty of the work is using sketching for mixing global feature information rather than dimensionality reduction (based on discussion with reviewers hHzn and 5P6B).
- We have added an expanded comparison table contrasting SRF with existing encoding approaches (based on discussion with reviewer hHzn). This table complements our discussion on how SRF overcomes weaknesses of existing methods.
- We have added practical hyperparameter selection guidelines in Appendix E (per the suggestion of reviewer hHzn).
- We have added additional notation needed to better understand Algorithm 1 to the Introduction (per the suggestion of Reviewer i8dd).
- We have added a block diagram visualizing Algorithm 1 in Appendix A (based on the suggestion of reviewer hHzn).

---

### Decision · Program_Chairs · 2025-09-17

**Decision:**

Accept (poster)

**Comment:**

This paper proposes Sketched Random Features (SRF): randomized global embeddings of node features that are injected at each GNN layer to mitigate oversquashing, oversmoothing, and limited expressivity. Reviewers engaged seriously and are broadly positive: two accepts and two borderline-accepts after rebuttal. They highlighted clear writing, modularity, solid theory, and consistent empirical gains, in particular on long-range benchmarks.

Concerns centered on perceived incremental novelty and comprehensiveness of evaluation breadth (diversity, ablations, baselines). The rebuttal substantially improved the case. The authors clarified the novelty—using sketching to mix global feature information rather than for dimensionality reduction, added a contrast table situating SRF among positional/random-feature encodings, provided hyperparameter selection guidelines and additional ablations, included additional backbones, and discussed inductive deployment and efficiency.

In the light of these discussions, I find SRF to be a simple, task-agnostic “add-on” with solid theoretical support, consistent gains, and favorable complexity. I hope that the remaining points can indeed be addressed at the camera-ready stage as promised.